# Hyperpolarized Xenon-129 Chemical Exchange Saturation Transfer (HyperCEST) Molecular Imaging: Achievements and Future Challenges

**DOI:** 10.3390/ijms25031939

**Published:** 2024-02-05

**Authors:** Viktoriia Batarchuk, Yurii Shepelytskyi, Vira Grynko, Antal Halen Kovacs, Aaron Hodgson, Karla Rodriguez, Ruba Aldossary, Tanu Talwar, Carson Hasselbrink, Iulian C. Ruset, Brenton DeBoef, Mitchell S. Albert

**Affiliations:** 1Chemistry Department, Lakehead University, Thunder Bay, ON P7B 5E1, Canada; vbatarch@lakeheadu.ca (V.B.);; 2Thunder Bay Regional Health Research Institute, Thunder Bay, ON P7B 6V4, Canada; 3Chemistry and Materials Science Program, Lakehead University, Thunder Bay, ON P7B 5E1, Canada; 4Applied Life Science Program, Lakehead University, Thunder Bay, ON P7B 5E1, Canada; 5Physics Program, Lakehead University, Thunder Bay, ON P7B 5E1, Canada; 6Chemistry & Biochemistry Department, California Polytechnic State University, San Luis Obispo, CA 93407-005, USA; 7Xemed LCC, Durham, NH 03824, USA; 8Department of Chemistry, University of Rhode Island, Kingston, RI 02881, USA; 9Faculty of Medical Sciences, Northern Ontario School of Medicine, Thunder Bay, ON P7B 5E1, Canada

**Keywords:** molecular imaging, xenon, NMR, hyperpolarized MRI, chemical exchange saturation transfer, biosensors, HyperCEST, personalized medicine

## Abstract

Molecular magnetic resonance imaging (MRI) is an emerging field that is set to revolutionize our perspective of disease diagnosis, treatment efficacy monitoring, and precision medicine in full concordance with personalized medicine. A wide range of hyperpolarized (HP) ^129^Xe biosensors have been recently developed, demonstrating their potential applications in molecular settings, and achieving notable success within *in vitro* studies. The favorable nuclear magnetic resonance properties of ^129^Xe, coupled with its non-toxic nature, high solubility in biological tissues, and capacity to dissolve in blood and diffuse across membranes, highlight its superior role for applications in molecular MRI settings. The incorporation of reporters that combine signal enhancement from both hyperpolarized ^129^Xe and chemical exchange saturation transfer holds the potential to address the primary limitation of low sensitivity observed in conventional MRI. This review provides a summary of the various applications of HP ^129^Xe biosensors developed over the last decade, specifically highlighting their use in MRI. Moreover, this paper addresses the evolution of *in vivo* applications of HP ^129^Xe, discussing its potential transition into clinical settings.

## 1. Introduction

Molecular imaging is an evolving field within medical imaging and is defined as the visualization, characterization, and measurement of biological processes at the molecular and cellular levels [1]. It has the potential to become a stepping stone for active implementation and further development of personalized medicine paradigms. The advantages of molecular imaging hold great promise for precision oncology [2,3], early detection and treatment monitoring of neurodegenerative diseases [4,5], and detection, imaging, and evaluation of inflammation [6,7]. Currently, medical imaging modalities capable of molecular imaging include positron emission tomography (PET), single-photon emission computed tomography (SPECT), photoacoustic imaging (PAI), and, recently, magnetic resonance imaging (MRI). While traditional molecular imaging modalities such as PET and SPECT have superior sensitivity and can detect minimal concentrations of contrast agents or molecular targets, they have some severe disadvantages. First and foremost, the spatial resolution of PET and SPECT is substantially limited [8], although there have been significant improvements in the field of organ-specific PET [9,10]. Both these imaging modalities rely on the utilization of radioactive tracers and, therefore, require fast imaging directly after synthesis and purification of the radiolabeled molecular imaging agent due to their respective short half-life time. Although relying on ultrasonic wave propagation instead of ionizing radiation, PAI also has limited spatial resolution due to the limited bandwidth, finite size of a transducer, and tissue-dependent propagation velocity of the ultrasonic waves [11].

Conventional MRI does not perform well in molecular imaging settings. Due to the high background signal and overall lack of sensitivity, conventional MRI contrast agents are not well suited for molecular imaging purposes. On the other hand, progress towards molecular imaging has been made with the extensive development of non-proton (multinuclear) MRI. This paradigm partially resolves the issue of strong background signal from water protons in the case of exogenous nuclei such as fluorine-19 (^19^F) or xenon-129 (^129^Xe). Multinuclear MRI is also suitable for metabolic imaging due to well-distinguished chemical shifts of numerous biomolecules [12,13,14], drugs [15,16,17], and their respective metabolites [18,19]. In addition, a wide range of advanced MRI techniques dedicated to improved sensitivity and signal augmentation are available in the field of non-proton MRI.

The most notable and prominent technique that provides a up to five orders of magnitude signal boost is hyperpolarization [20,21]. This creates a metastable nuclear spin-polarized (hyperpolarized; HP) state, resulting in a substantial increase in nuclear magnetization. It is possible to hyperpolarize carbon-13 (^13^C), fluorine-19 (^19^F), and noble gases such as ^129^Xe [20,22,23] and helium-3 (^3^He) [24,25] using a dynamic nuclear polarization [18,26] and a spin-exchange optical pumping [23,27] techniques, respectively. While the increase in the MRI signal due to HP is substantial and on its own allows the conduction of molecular and metabolic imaging, it is possible to further increase its sensitivity by several orders of magnitude in the case of dissolved-phase HP ^129^Xe MRI. In order to achieve this, supramolecular macrocycles that are capable of reversibly binding ^129^Xe can be implemented as contrast agents. In 2006, Schröder et al. [28] introduced a pioneering method known as hyperpolarized chemical exchange saturation transfer (HyperCEST), which provides an extra sensitivity boost of up to three orders of magnitude for HP ^129^Xe MRI signal.

The HyperCEST contrast mechanism allows for a second wind into molecular MRI imaging, bringing hope that high-resolution and high-sensitivity imaging of molecular targets using MRI is possible and can be utilized for personalized medicine in the future. Despite extensive developments in the field of HyperCEST molecular sensing over the past two decades, there has been no demonstration of any disease detection using the HyperCEST effect yet. Our review paper aims to highlight the current progress and development in the field of HyperCEST molecular MRI, as well as discuss the technical challenges that should be overcome for further preclinical and clinical translation of this prominent imaging approach.

## 2. Hyperpolarized ^129^Xe MRI

In the context of molecular MRI, the prospect of developing a contrast agent that addresses the limitations of conventional MRI and offers enhanced sensitivity is highly promising for potential clinical applications. Xenon is a noble gas with a van der Waals radius of 2.2 Å. It is non-toxic, highly soluble in biological tissues, capable of dissolving in blood, and can diffuse across membranes. Its stable isotope, ^129^Xe (with a natural abundance of 26.4%), exhibits favorable nuclear magnetic resonance properties, making it suitable for molecular imaging. It has a gyromagnetic ratio of 11.77 MHz/T [29], spin of ½, no quadrupole moment, and a wide chemical shift range, allowing it to be used as a versatile contrast agent. Its solubility in aqueous solutions and affinity for hydrophobic binding pockets open avenues for functionalization by combining it with host structures that bind one or multiple gas atoms [30]. Its extreme sensitivity to the molecular environment, due to its large electron cloud, is especially essential in molecular imaging settings.

Significant progress has been made in the field since the introduction of ^129^Xe MRI in 1994 by Albert et al. [20] and its first application in humans was published by Mugler et al. [31]. The primary focus has been on applications in lung studies [29,32,33,34,35,36,37,38,39,40], which utilize a direct visualization of the inhaled xenon gas. In 2023, Polarean Imaging Plc reached a significant milestone in a momentous stride toward the clinical application of HP ^129^Xe MRI with the FDA approval of their XENOVIEW system [41]. It is the first and only inhaled MRI hyperpolarized contrast agent for assessing lung ventilation in adults and pediatric patients aged 12 years or older, broadening the scope of HP ^129^Xe MRI for pulmonary medicine. Due to sufficiently long *T*_1_ relaxation time in the blood [42], HP ^129^Xe has the potential to produce functional images of highly perfused organs, such as the brain [43,44,45,46,47,48,49,50] and kidneys [51,52]. In recent years, significant progress has been made in the development of potential applications for using ^129^Xe in a combination of targeted biosensor imaging. HP ^129^Xe MRI presents a notable advancement over conventional MRI, primarily attributed to the process of hyperpolarization.

### 2.1. Spin-Exchange Optical Pumping

Hyperpolarization in magnetic resonance provides significant improvements in signal intensity [20,53,54], offering 10^3^–10^5^-fold higher signal enhancement compared to thermal polarization. It provides superior contrast and sensitivity in imaging, making HP ^129^Xe MRI a valuable tool for the early detection and characterization of various physiological processes and pathological conditions. There are various hyperpolarization techniques [55] for solids, liquids, and gases that have been studied, including Brute Force Polarization (BFP) [56], Spin-Exchange Optical Pumping (SEOP) [57], Dynamic Nuclear Polarization (DNP) [58], Chemically Induced Dynamic Nuclear Polarization (CIDNP) and photo-CIDNP [59], Parahydrogen Induced Polarization (PHIP) [60], and Signal Amplification by Reversible Exchange (SABRE) [61]. The most common technique for obtaining HP ^129^Xe is SEOP [23,24]. This technique has its origins in the late 1970s [62,63] and entails a two-step physical process. First, circularly polarized laser light is employed to optically pump electrons within an alkali metal vapor (typically rubidium or cesium) with a single electron in its outer shell. Second, spin-exchange collisions take place between the optically pumped alkali metal vapor and the noble gas atoms, such as ^129^Xe. These collisions facilitate the transfer of angular momentum, leading to the enhanced nuclear spin polarization of the noble gas.

An SEOP cell contains vaporized rubidium (Rb), xenon (Xe), and nitrogen (N_2_) within an external magnetic field, termed B_0_. The simplified polarization setup is shown in Figure 1A. If the laser is tuned in resonance with the D_1_ transition (794.7 nm for Rb), the circularly polarized photons from the laser beam excite the alkali metal electrons from state 5 ^2^S_1/2_ to 5 ^2^P_1/2_. According to the Zeeman effect, each of the two hyperfine states, 5 ^2^S_1/2_ to 5 ^2^P_1/2_, undergo splitting into two sublevels: m_J_ = −½ (spin-down) and m_J_ = +½ (spin-up), where J is a total angular momentum quantum number. The selection rule Δm_J_ = ±1 applied for transitions induced by polarized light depends on right or left circularity. Left-circularly polarized light, therefore, selectively excites Rb valence electrons from the m_J_ = −½ level in the ground state (5 ^2^S_1/2_) to the m_J_ = +½ level in the excited state (5 ^2^P_1/2_). Sustained excitation through optical pumping increases the population of one of the two sublevels, resulting in a non-zero net population difference. The interaction of Rb atoms with left-circularly polarized light is shown in Figure 1B. In spin-exchange optical pumping, several simplifying conditions are commonly applied [62]. Firstly, pressure broadening of the absorption line is prevalent, making the alkali-metal hyperfine structure unresolved due to high total noble gas and nitrogen densities. Secondly, the presence of a quenching gas (N_2_) eliminates radiation trapping as a relaxation source. Spin relaxation of the rubidium electrons occurs through non-radiative (Rb-Rb, Rb-N_2_, Rb-Xe and cell-wall) and radiative interactions, where the alkali metal atoms emit randomly polarized light, destroying spin polarization [64]. In the second stage of the SEOP process, which is designed to produce hyperpolarized xenon gas, the effective transfer of hyperpolarization from optically pumped rubidium to the xenon nucleus is crucial (Figure 1C). In the gaseous phase, the rapid motion results in the averaging of direct dipolar interactions to zero. In contrast, the Fermi contact interaction becomes significant when noble xenon gas is in close proximity to rubidium atoms. This spin-exchange process may be mediated by two-body (Rb-Xe) or three-body (Rb-Xe-N_2_) collisions [65]. In the context of a three-body interaction, Fermi contact becomes more effective as energy from the collision dissipates into a third atom. This process may result in the temporary formation of van der Waals complexes, which persist until another collision with a different body leads to their disintegration. The presence of a high concentration of Xe reduces the three-body exchange rate due to increased collision rates between the van der Waals complexes, which is detrimental to achieving polarization. Thus, as was demonstrated by Six et al. [66], the noble gas is kept at pressures ranging from 5 kPa to 200 kPa and is usually used in a mixture, usually in either a ^4^He-N_2_ mixture or with pure N_2_ gas. The hyperpolarization of xenon nuclei starts to decay once they are no longer able to collide with spin-polarized alkali metal atoms. Consequently, it is essential to promptly transfer and utilize the noble gas before a significant loss of hyperpolarization occurs.

While the considerable cost of commercial hyperpolarization equipment has restricted its widespread use in clinical settings, the advantages of HP ^129^Xe MRI have driven ongoing advancements in production rates and manufacturing. Two types of systems are commonly used to produce hyperpolarized ^129^Xe—a flowing cell or continuous-flow [67,68,69,70,71] design and static cell or stopped-flow systems [66,72,73]. Flowing cell polarizers provide higher production rates due to continuous gas circulation, while static cell polarizers achieve high polarization levels over a longer time without the need for additional cooling. While static cell polarizers have reported higher ^129^Xe polarization values, its xenon production rates typically range around 100 mL/h, with the highest observed at ~1000 mL/h. On the other hand, continuous-flowing cell methods exhibit production rates exceeding 1000 mL/h [74]. Presently, the primary producers of ^129^Xe hyperpolarization systems on the market include Polarean Imaging Plc (flowing cell design), XeUS Technologies Ltd. (static cell design), and Xemed LLC (flowing cell design) [75]. The choice between designs depends on the specific application requirements, balancing the need for rapid production against achieving and maintaining high polarization levels.

### 2.2. HP ^129^Xe Chemical Exchange Saturation Transfer

The progress in clinical MRI applications has been significantly influenced by the development of spin hyperpolarization techniques. Given that ^129^Xe is an exogenous agent, it should be introduced into the area of interest and remain detectable at relatively low concentrations. Its restricted solubility in aqueous solutions (and consequently in many biological tissues) is dictated by an Ostwald solubility coefficient of approximately 3.3 mM/atm [53]. Thus, the low spin density of xenon poses a challenge for direct detection in numerous applications.

Chemical exchange saturation transfer (CEST), introduced by Ward et al. [76], is a rapidly evolving molecular MRI technique that relies on various reporters and is typically employed to indirectly detect contrast agents following magnetization transfer from selectively excited exchangeable sites [77,78,79,80,81]. The fundamental principles of CEST [82] are based on the saturation transfer from the solute protons to the bulk protons engaged in the chemical exchange process [83]. Initially, solute protons experience saturation through the application of a selective radiofrequency (RF) pulse at the resonant frequency of the solute. Due to chemical exchange, this saturation is then transferred to the surrounding water protons. Considering that the water pool is significantly larger than the solute pool, each exchanging saturated proton is replaced successively by a non-saturated water proton, which undergoes subsequent saturation. Prolonged irradiation, under conditions of sufficiently fast exchange rates and a long saturation time, leads to a substantial enhancement of the saturation effect and a signal reduction in the larger water pool. This indirect method, therefore, enables the imaging of low-concentration solutes and is visualized by using the Z-spectrum, which plots the remaining z-magnetization after each saturation offset as a function of saturation frequency [84].

Hyperpolarized CEST (HyperCEST) is a unique approach that utilizes hyperpolarized nuclei, such as ^129^Xe, as a source of a signal [28]. Considering that xenon is not inherently a part of any biologically relevant molecules, the specificity of the MRI signal is established indirectly. This is achieved by encapsulating the noble gas within a molecular host that alters the resonance frequency of the hyperpolarized nuclei. HyperCEST relies on the reversible binding of ^129^Xe to a specific supramolecular cage, thus creating a unique chemical shift of the bulk (pool A) and enclosed (pool B) xenon reservoirs. Two distinguishable ^129^Xe resonances, one associated with pool A and the other with pool B, are usually well separated due to a significant chemical shift range. This enables their selective targeting through the use of specific RF pulses. When the RF pulse matches the resonance frequency (on-resonance) of the ^129^Xe within the cage, the nuclei undergo depolarization. Chemical exchange prompts depolarized atoms to enter the nearby dissolved pool, causing a decrease in signal (Figure 2A). Repeating compartment-selective depolarization, it is possible to completely deplete the MRI signal in close proximity to the supramolecular cage rendering it visible on the MRI scan. The difference between on-resonance and off-resonance decay, weighted by the saturation time, can be quantified as a saturation contrast. Hyperpolarized xenon exhibits a significantly prolonged relaxation time compared to protons [74], resulting in not only a more robust, but also a longer-lasting, enhanced MRI signal. Therefore, such an approach combines saturation transfer enhancement and the utilization of spin hyperpolarization, resulting in an additional sensitivity boost.

The concept of ^129^Xe biosensors was first introduced in 2001 [85]. Following this, efforts have been focused on creating and enhancing various macromolecular cages, along with their functionalization, as we will delve into in more detail in this review. The necessity to evaluate the characteristic parameters of these proposed molecular systems and facilitate comparisons across different setups has led to the development of a quantitative HyperCEST theory. The changes in the macroscopic magnetization of a nuclear species undergoing exchange between pools A and B under a saturation pulse are described by Bloch–McConnel equations [86]. The saturation pulse, characterized by its power (B_1_) and duration (t_sat_), induces a HyperCEST signal loss. Moreover, an intrinsic depolarization of HP ^129^Xe accelerates this loss. Thus, a full HyperCEST (FHC) solution [87] was proposed as a simplified analytical solution of the Bloch–McConnel equations, which considers both depolarization mechanisms. This model is usually used to evaluate main system parameters, such as the binding constant (k_A_), host occupancy (β), the ratio of bound and free xenon (f_B_), the exchange rate between two pools (k_AB_), and the longitudinal and transverse relaxation times of bulk ^129^Xe (*T*_1_*^A^*, *T*_2_**^A^*). Kunth et al. [88] have demonstrated that biosensor characterization can be accomplished by modulating the saturation pulse and analyzing the obtained z-spectra. They experimentally determined the exchange parameters for the cryptophane-A monoacid in dimethyl sulfoxide and showed that acquiring multiple HyperCEST z-spectra with varying saturation pulse strengths enables accurate and simultaneous quantification of multiple parameters. This approach has been further developed in recent years and has been used in various applications [30,74,89,90,91,92], which we will partially consider in this review.

HyperCEST is crucial for its ability to enhance the sensitivity and specificity of MRI, particularly in molecular imaging applications. By exploiting the chemical exchange between hyperpolarized contrast agents and target molecules, HyperCEST allows for highly sensitive detection of specific molecular interactions.

### 2.3. Hardware and Pulse Sequence Considerations

While the vast majority of HyperCEST research has been conducted on nuclear magnetic resonance (NMR) spectrometers [93,94,95,96,97,98], only a handful of studies have utilized MRI scanners [99,100,101,102,103,104]. Furthermore, only two groups have conducted HyperCEST experiments at clinically relevant magnetic field strengths of 2.89 T [104] and 3.0 T [99,100,101,103]. Further increases in the number of studies performed using MRI scanners and, especially, clinical whole-body scanners are of paramount importance for facilitating the translation of HyperCEST molecular imaging to potential preclinical and clinical studies.

There are multiple fundamental and essential differences between NMR spectrometers and MRI scanners that should be taken into consideration, and which will determine the practical feasibility of the potential molecular imaging agent. First and foremost, the magnetic field strength is superior in NMR spectrometers due to the overall small size of their bore. Such a strong magnetic field results in a superior spectral resolution and signal-to-noise ratio (SNR) allowing for the achievement of a high level of HyperCEST effect with relative ease. In addition, the location of small NMR detector coils in close proximity to the NMR sample results in a high filling factor, further improving the achievable SNR. Finally, the small size of the NMR bore allows for precise shimming of the external magnetic field (resulting in linewidth of ~3–4 Hz) [105], ensuring the highest level of ^129^Xe depolarization effectiveness.

On the contrary, MRI scanners are operating at much lower magnetic fields. Clinical MRI machines rely on a magnetic field strength of 3.0 T and lower. The small-animal scanners can go up to 9.4 T. In conventional MRI, enhancing the strength of the magnetic field, B_0_, offers advantages through the subsequent rise in equilibrium magnetization and the increased energy of the transition. This leads to a signal scaling proportional to the (B_0_)^2^. In HP ^129^Xe MRI, however, the signal is determined by the hyperpolarization technique; therefore, the magnitude of the hyperpolarized magnetization is field independent. Since the SNR(HP)/SNR(water) ratio scales as 1/B_0_, HP ^129^Xe MRI has a decisive advantage over conventional MRI at lower fields [106,107]. The acceptable B_0_ shim level decreases with an increase in the MRI magnetic field from 7–20 Hz linewidth for 1.5 T to 13–60 Hz linewidth for 7.0 T [108]. Each step of the B_0_ shimming process involving measurements, analysis, and adjustment is susceptible to specific artifacts, sub-optimal choices, and errors resulting in the overall complexity of the B_0_ shimming. On most MRI scanners, the selection of the shim volume of interest is purely geometry-based from MR images without consideration of the underlying phase behavior [108]. This challenge may result in a HyperCEST imaging quality decrease while performing *in vivo* HyperCEST imaging or *in vitro* imaging of large inhomogeneous phantoms.

Since all MRI scanners are equipped with a built-in proton body coil, conducting HyperCEST experiments requires the utilization of additional ^129^Xe MRI coils. While there are several types of MR coils available, currently, the dual-tuned ^1^H/^129^Xe volumetric birdcage coils have been used for all MRI HyperCEST studies [100,101,102,103,109]. The birdcage coil is considered to be the most successful for MRI applications due to its strong B_1_ magnetic field homogeneity and achievable high SNR [110]. While these advantages are sufficient for initial MRI HyperCEST evaluation and imaging of the HyperCEST imaging agents *in vitro* and in small animals, considerations towards multichannel receivers may be useful for further improvement of HyperCEST imaging performance. HP ^129^Xe dissolved-phase imaging has an overall low SNR, resulting in limited spatial resolution (mostly of 32 × 32 acquisition matrix). Recently, the implementation of a multichannel RF receiver coil array demonstrated an HP ^129^Xe dissolved-phase SNR increase in human subjects [111]. This increase in the SNR may potentially lead to the acquisition of HP ^129^Xe with higher spatial resolution resulting in better spatial resolution of the HyperCEST images. This potential resolution gain may debase the dominance of the birdcage coils in favor of phased arrays for future HyperCEST molecular imaging.

Unlike NMR spectrometers, MRI scanners are incapable of continuous depolarization of HP ^129^Xe nuclei, which is commonly used for HyperCEST experiments [93,95,112,113]. Instead, the application of a limited number of depolarization pulses of various shapes can be implemented in MRI machines. This difference originates from safety considerations for specific absorption rates (SARs) [114]. SAR becomes a main limiting factor in clinical MRI systems and prevents the application of a high number of depolarization pulses or RF pulses with high intensity. Therefore, it is of paramount importance to optimize the depolarization pulse train by carefully considering the pulse magnitude, pulse shape, phase, total number of RF pulses, and the time interval between pulses. Surprisingly, the first HyperCEST MRI pulse sequence parameter optimization was performed recently in 2023 by Grynko et al. [103] for cucurbit[6]uril using a 3.0 T clinical MRI scanner. Among the evaluated pulse shapes, a 3-lobe sinc pulse demonstrated the best performance due to its relatively large bandwidth and high transmitted RF energy. Although the authors studied the effect of multiple RF pulse shapes and pulse angles on the HyperCEST performance of the cucurbit[6]uril molecule, they did not look into the potential effect of RF depolarization phases and interpulse time delays on the HyperCEST effect. While Grynko et al. performed RF depolarization pulse optimization thoroughly using a single-voxel MRS pulse sequence, a similar study is required to assess the performance and optimization of the RF pulses for HyperCEST imaging with meticulous SAR and B_1_ inhomogeneity considerations. Furthermore, evaluating more complex pulse shapes and their phases may, and likely will, be of high interest for practical *in vivo* HyperCEST molecular imaging.

In addition, multiple non-Cartesian k-space trajectories should be considered for HyperCEST molecular MRI imaging. The implementation of spiral [115,116] and radial [117,118] k-space sampling may be beneficial for HyperCEST MRI due to the minimization of *T*_1_ and *T*_2_*** effects and, therefore, the relaxation influence on the final HyperCEST MRI image. Finally, a k-space correction for the excitation flip angle [119,120] and for RF inhomogeniety [117] should be performed before HP ^129^Xe off-resonance and on-resonance image reconstruction to improve the accuracy of the localized HyperCEST effect quantification. Moreover, implementation of off-resonance and on-resonance image denoising using advanced image post-processing algorithms [121,122,123] (instead of simple thresholding that is currently used [99,100,102]) will substantially improve the final quality of the HyperCEST image and prevent any subtraction artifacts from appearing.

## 3. HyperCEST Active Agents

In the realm of HyperCEST molecular imaging, the choice and design of biosensor systems play a pivotal role in achieving optimal contrast and sensitivity. An efficient xenon biosensor should meet specific criteria, including a high affinity for xenon, facilitating rapid in–out exchange for efficient replenishment with unbounded polarized xenon while maintaining a slow exchange on the xenon chemical shift time scale. Minimizing relaxation for encapsulated xenon is crucial, and in a multiplexing approach, distinct chemical shifts for xenon in different host molecules are required. Furthermore, a significant resonance frequency difference between signals from xenon in blood and within the host that enhances selective excitation at the caged xenon frequency is beneficial for *in vivo* applications. Given the hydrophobic nature of xenon and its van der Waals radius of 2.2 Å, it is important to incorporate an optimal ratio of 0.55 [124] between the size of the guest and the host cavity (the optimal cavity volume is ~80 Å) when designing the host. Combined with an affinity tag for a particular biomarker, the xenon@host molecule can function as a biosensor. When the HP ^129^Xe combines with various molecular cages, such as cryptophane-A [125], cucurbit[6]uril [104], proteins [126], gas vesicles [127], and metal–organic frameworks [128], it holds significant potential for applications in molecular imaging and the detection of trace molecules within living structures. The recent review by Jayapaul and Schröder [89] provided a concise overview of synthesizing and handling HP ^129^Xe, addressing the challenges in production, host structure strategies, and functionalization. Thus, we will not delve into these aspects but will explore the application of the diverse host structures suitable for designing ^129^Xe biosensors within the context of HyperCEST molecular MRI.

### 3.1. Cryptophane-A

#### 3.1.1. Cryptophane-A Structure

Among all xenon biosensors, cryptophanes [129,130,131,132,133,134,135,136,137] stand out as the most extensively studied host molecules. In the 1980s, Collet [138] made significant contributions by synthesizing a novel class of molecules known as cryptophanes. Cryptophane is defined by two 3-fold symmetric cyclotribenzylene units (CTBs), often called cyclotriveratrylenes (CTVs), connected by linkers [139]. It exists in a concave “crown” conformation, which sustains a relatively shape-persistent internal cavity. Without encountering any steric constraints, a guest molecule can freely enter and exit the hydrophobic cavity. The form and properties of cryptophanes exhibit considerable variation that is influenced by factors such as stereochemistry, the type of linker unit connecting the CTV bowls, and the functional groups attached to the CTV macrocycles. Given that the guest-binding strengths and preferences of cryptophanes are primarily influenced by the dimensions of their internal cavities, which are determined by the length of the linkers connecting the CTV, it is more practical to characterize the cryptophane “core” based on its linkers. Most cryptophanes exhibit two diastereomeric forms(syn- and anti-)distinguished by their symmetry type. The anti-cryptophane isomer belongs to the D_3_ point group and the syn-cryptophane isomer belongs to the C_3h_ point group [140]. The CTV components are known for their flexibility, allowing them to invert or take on a saddle-twist conformation at room temperature [141]. This gives rise to four potential cryptophane conformers, including out–out, in–out, in–in, and out–saddle conformers.

Cryptophanes are categorized based on nomenclature representing the number of carbons available at the linker [140]. The majority of cryptophane derivatives employ alkyldioxy linkers (–O(CH_2_)_n_O–) or more generally organodioxy linkers (–OC_n_O–) [139]. Thus, it is possible to indicate the size of the “core” cryptophane cavity by reference to the number of carbon atoms (n,m,l) in each of its three linkers. The initial cryptophanes were named in alphabetical order, based on the order in which they were synthesized. The first cryptophane-A (Figure 3A, adapted with permission from Bartik et al. (1998) [142]) was synthesized in 1981 [143,144] via the linkage of two CTV caps by three ethylene linkers, with a volume of the internal cavity of V_cav_ ~ 87–119 Å. Cryptophane-A (CrA) is characterized by a high binding constant (k_A_ = 3900 M^−1^ at 278 K in 1,1,2,2-tetrachloroethane-d_2_) [145] and a distinctive ^129^Xe NMR signal that is well differentiated for both cryptophane-bound (in the range of 30 to 80 ppm) [146] and dissolved xenon (Figure 3B) [147]. Significant advancements have been achieved in synthesizing novel cryptophane derivatives, characterizing their structural aspects, and exploring their distinctive host@guest chemistry. These developments have been comprehensively covered in existing reviews [30,147,148]. Our focus will primarily be on CrA, highlighting the advancements made over the past decade.

#### 3.1.2. Cryptophane-A Exchange Kinetics Properties

To comprehend the intricate dynamics of interactions between host and guest molecules in various applications, it is crucial to understand their kinetic behavior. This involves assessing essential parameters such as the exchange rate, binding constant, host occupancy, chemical shift, and both transverse and longitudinal relaxation times. The dynamics of the ^129^Xe@host structures depend on the mechanism of binding and the associated energy barrier, structural modifications of the CrA, ^129^Xe concentrations in the solvent, the solvent, and the environmental conditions (temperature, pH, etc.). It has been shown [149] that xenon encapsulation by CrA is mainly driven by hydrophobic and non-covalent interactions. It is characterized by the association constant K_A_, which can be calculated as K_A_ = (1/c_0_) × exp (−ΔA_bind_/RT) (where c_0_—const; R—the universal gas constant; T—absolute temperature; and ΔA_bind_—binding free energy). ^129^Xe@CryA (including some derivatives) are kinetically stabilized by constructive binding with the respective binding energy in the range of 4–17 kJ mol^−1^ [134]. It is not required to overcome the large steric constraints of CrA; therefore, xenon residence times are on the order of milliseconds [142]. CrA can have various conformations, which will affect the size of the cavity and the stability of the Xe@CrA system. As previously stated, Xe@CrA demonstrated a binding constant of K_A_~3900 M^−1^ and a chemical shift of ~62 ppm [147] at 278 K in C_2_D_2_Cl_4_. In the HyperCEST settings, another parameter that plays a critical role in evaluating Xe@CrA kinetics is the exchange rate. The exchange rate is usually described by an Eyring–Polanyi equation [150]: K_EP_ = (k_B_T/h) × exp(−ΔA_bind_/RT) (where k_B_ is a Boltzmann constant and h is the Planck constant). There are two different types of exchange mechanisms for a Xe@CrA system—dissociative and associative (depending on the host and xenon concentrations). According to Hauk et al. [151], the dynamics between guest molecules and container molecules reveals that gating has a critical influence on the ease of formation and stability of host–guest complexes. These mechanisms are referred to as the “French door” and “sliding door”. In the “French door”, the exchange is facilitated by a significant conformational change, resulting in the opening of host portals and promoting guest exchange. Alternatively, the “sliding door” mechanism involves a process akin to guiding guests through flexible portals by adjusting the conformation of linkers, similar to squeezing them through. Noticeable variations in xenon release rates from diverse hosts and in various solvents are identifiable by examining alterations in the observable effective transverse relaxation time and can be compared using Swift–Connick plots [91].

The exchange kinetics depend on the properties of the solvent, such as its composition, polarity, and temperature. It was shown [152] that the Xe@cryptophane-A monoacid (CrA-ma) in two different solvents (DMSO and H_2_O) showed entirely different exchange kinetics for a given temperature. However, many of the studies do not include explicit solvent effects. Hilla and Vaara [134] conducted computational analyses to investigate the interactions between xenon and CrA in its native solvent environment at room temperature, with a primary focus on determining the binding free energy (ΔA_bind_) and understanding the mechanism of xenon complexation. Molecular dynamics (MD) and metadynamics (MTD) simulations were performed at the semi-empirical GFN2 and GFN0, as well as at the GFN-FF force-field levels of theory. The ΔA_bind_ was estimated by a thermodynamic cycle for four distinct physical sites (water, xenon atom in water, CrA in water, and Xe@CrA in water). The obtained values correctly reproduced the negative sign of ΔA_bind_, which were in good agreement with prior experiments (5–6 kcal mol^−1^). Using the obtained ΔA_bind_, both the association constant (K_A_) and Eyring–Polanyi exchange rate (K_EP_) were calculated between Xe@CrA and Xe in solution. For GFN-FF, GFN0, and GFN2, K_A_ = 1.6 × 10^3^, 2.8 × 10^7^, and 2.7 × 10^15^ M^−1^, as well as K_EP_ = 3.8 × 10^9^, 2.2 × 10^5^, and 2.4 × 10^−3^ s^−1^, respectively, were obtained. K_A_ resulting from GFN-FF had the best agreement with the existing values (ca. 10^3^–10^4^ M^−1^) and it was found that GFN0 produces the closest order of magnitude for K_EP_ (ca. 10^2^s^−1^). The simulation of the dissociation of Xe atoms out from the CrA cavity was performed by using MTD. MTD simulations enabled the complexation mechanism and pathway of Xe to be identified for the first time. In the “French door” part, the two OCH_3_ groups rotate away from the Xe dissociation pathway. In the “sliding door” part, the entire host structure opens by stretching the linkers that connect the CTV bowls. The mean residence time (t_r_) of water molecules, measured using approximation through a single-exponential fit, resulted in values of 198 and 308 ps at the GFN0 and GFN-FF levels, respectively.

#### 3.1.3. HyperCEST Approaches Utilizing CrA-Based Agents

Consequently, the performance of a HyperCEST agent is shaped by its molecular environment. An important feature of Xe@host complexes is that each of them has a unique exchange kinetic fingerprint (k_A_, Δω, β, f_B_, k_AB_, k_BA_, *T*_1_*^A^*, and *T*_2_**^A^*). In 2022, Kunth et al. [92] took a significant stride in advancing the process of exchange kinetic fingerprinting through the utilization of the quantitative HyperCEST approach. Fast echo planar imaging (EPI)-based HyperCEST MRI in combination with Bloch–McConnell analyses was performed in order to map the unique fingerprint of a CrA-ma. The quantification method was applied to CrA-ma in DMSO for a two-compartment sample with identical CrA-ma molecules at two different concentrations. The outer compartment contained the known reference concentration (50 μM), whereas the inner compartment carried the “unknown” concentration. A pixel-wise qHyperCEST analysis was performed for saturation with B_1_ of 3.3; 5.6; 2.2 µT and t_sat_ of 5; 10; 15 s. Pixel–pixel signal fitting was performed using the FHC solution. Each pixel carries quantitative information regarding the ratio of bound and free Xe (f_B_), the Xe exchange rate (K_BA_), the relative chemical shift (Δδ), and the total fitting times per pixel. All parameters were in excellent agreement with previously reported values derived from ROI-averaged data: chemical shift difference Δδ = (166.69 ± 0.02) ppm, the exchange rate K_BA_ = (290 ± 20) s^−^^1^, and the host occupancy β = 9%. Quantitative xenon host concentration determination was achieved by utilizing an internal standard with a known host concentration, enabling the characterization of samples with unknown host concentrations based on identical exchange kinetics; this was facilitated by exploiting the constant nature of both host occupancy (β) and the affinity constant (K_A_) in DMSO over a CrA-ma concentration range, and the resulting histogram analysis revealed two populations around 25.5 ± 0.5 µM and 49 ± 2 µM (Figure 4). Therefore, the quantitative exchange kinetics mapping method for ^129^Xe hosts based on qHyperCEST that can disentangle the bound xenon fraction and the exchange rate (which are usually represented in intensity as a product term) has been demonstrated.

The initial demonstration of a ^129^Xe biosensor occurred in 2001 when Spence et al. [85] utilized CrA with an attached polar peptide chain to detect biotin–avidin binding. Generally, the functionalization of CrA with peptidic units improves its solubility in water. Significantly, a diverse range of applications involving CrA and derivatives based on cryptophane were carried out, focusing on the targeting of biological receptors or cells. Riggle et al. [153] presented a “smart” ^129^Xe NMR biosensor that underwent a peptide conformational change and labeled cells in acidic environments. By attaching a pH-responsive, membrane-inserting peptide and two water-solubilizing moieties to a tripropargyl cryptophane host, they were able to generate an ultrasensitive ^129^Xe NMR biosensor capable of labeling cells in an acidic microenvironment. The development of a cryptophane–EALA peptide conjugate, designed for membrane insertion at acidic pHs, represented progress towards the goal of creating highly sensitive ^129^Xe MR contrast agents for cancer diagnosis and treatment. A HyperCEST effect was observed with a 34 pM concentration, showcasing a sensitivity 8–9 orders of magnitude higher than that of conventional MR contrast agents. Following this, Zeng et al. [154] developed a new biosensor to detect biothiols, which consists of a CrA cage, a disulfide linker, and a triphenylphosphonium-functionalized naphthalimide group. This biosensor showed good stability across a broad range of pH values and proved its usability for the detection of biothiols. The detection threshold of these biosensors was found to be 200 pM [155]. Saturation contrast was evident exclusively in the biosensor incubated with the HCC827 cell line, characterized by high EGFR expression, and not in the biosensor incubated with the A549 cell line, which demonstrated intermediate EGFR expression.

The primary challenge in ^129^Xe@CrA systems is the restricted solubility of CrA. To address this issue, potential solutions include functionalizing the CrA host or developing new cryptophane-based cages through synthesis. In order to enhance the solubility of CrA biosensors, water-soluble cryptophanes were synthesized based on the anti-cryptophane-A conformation. However, the synthesis of syn-cryptophane-B (the diastereomeric molecule of CrA) has been achieved [156]. Léonce et al. [135] performed a comparative analysis of a novel syn-cryptophane-222-hexacarboxylate host system with its anti-diastereomer and cucurbit[6]uril, revealing xenon binding constants and the introduction of xenon in–out exchange. They showed that the resonance frequency of xenon in anti-cryptophane-222 hexacarboxylate and in syn-cryptophane-222-hexacarboxylate differs by more than 50 ppm (at 11.7 T), which can allow interesting multiplexing experiments. The syn-cryptophane-222-hexacarboxylate outperformed its anti-diastereomer and various anti-cryptophanes, being exempt from forms unable to encapsulate xenon. Additionally, it does not complex dissolved oxygen, exhibits high xenon-binding constants, and displays faster xenon in–out exchange. Recent research by Clément et al. [136] demonstrated that the introduction of nitrogen in place of a crown methylene bridge enhances solubility in the organic media of both the cryptophane and the synthetic intermediates while presenting the same conformation as known cryptophanes. Taking advantage of the enhanced solubility of nitrogen-substituted intermediates, the synthesis of anti-10 was performed on a multigram scale. The hyperpolarized ^129^Xe NMR spectrum of the Xe@anti-10 complex was recorded in C_2_D_2_Cl_4_ at several temperatures from 278 to 253 K. The exchange rate of xenon at 278 K was found to be 371 s^−1^ (residence time inside the cavity of 1.3 ms) with an affinity constant of 4300 ± 1000 M^−1^ (comparable to cryptophane 3900 ± 500 M^−1^). Furthermore, the functionalization possibilities have been shown by alkylation with propargyl bromide (65% yield), which afforded the clickable compound 12, and by acylation with bromobutyryl chloride, which produced the electrophilic compound 13 at a 69% yield.

The broad chemical shift range of ^129^Xe NMR allows for the utilization of multiple hosts, enabling the examination of various analytes based on differences in observed chemical shifts. This approach, known as “multiplexing”, has been implemented in various applications of ^129^Xe HyperCEST NMR and MRI. The first demonstration was performed by Berthault et al. [157] and was later developed by Tyagi et al. [158] through using CrA functionalized with polyglycerol dendrons. The experiments highlight the potential for imaging different cell types using a multichannel detection approach, as the ^129^Xe NMR chemical shift of cell-internalized cryptophane remains nearly independent of cell type [159]. Klippel et al. [160] demonstrated a concept for multichannel MRI cell labeling. They proposed a sensitive MRI approach that allows for the multiplexed localization of two individual nanoparticulate xenon hosts that are acting as cell tracers: CrA and perfluoroctyl bromide (PFOB) nanodroplets. Both hosts confer a unique chemical shift to ^129^Xe to enable a frequency-selective detection of two distinctly labeled mammalian cell populations resulting in a switchable “two-color” MRI contrast. Therefore, the imaging of mouse fibroblast cells was achievable through a frequency-selective saturation transfer, where the observed contrast for the two hosts relied on the saturation frequency. This work illustrates the feasibility to perform highly sensitive, multichannel detection of mammalian cells through HP ^129^Xe MRI using synthetic xenon nanocarriers as contrast agents.

^129^Xe encapsulated within a CrA host exhibits unique NMR properties, allowing for precise detection and localization within biological systems. This targeted approach enhances the imaging capabilities, enabling the visualization of specific biological processes or structures with an improved SNR and contrast by utilizing HP ^129^Xe MRI. *In vitro* MRI studies utilizing the HyperCEST approach with CrA-based biosensors has been applied in various applications, including contrast agents with dual imaging capabilities (such as fluorescence imaging) [161], thermometry applications, and studies within living cells using the HyperCEST method. The pioneering work by Schröder et al. [159] was designed to verify the HyperCEST response of CrA-ma cells at physiological temperatures. Mouse fibroblasts (L929) were incubated for 3 h with 90 mm CrA-ma dissolved in cell culture medium before they were washed and resuspended at a medium density. An accumulation of 64 direct scans revealed four different xenon pools chemical shifts: Xe@cells at 196 ppm, Xe@solution at 192 ppm, Xe@CrA-ma at 59 ppm, and Xe@gas at 0 ppm. Utilizing CrA modified with fluorescein, researchers successfully employed ^129^Xe MRI to distinctly distinguish between labeled and non-labeled cells. This differentiation was achieved with an average intracellular CrA concentration of 15 μM, attained after incubating cells with 50 μM cryptophane for 20 h. In a later paper [162], their group was able to distinguish two types of cells using ^129^Xe MRI (9.4 T) with CrA. This was achieved through using a concentration of 20 nM by functionalizing CrA and fluorescein with biotin, conjugating an anti-CD14 antibody to avidin, and incubating these constructs with high-CD14-expressing macrophages and a control fibroblast cell line. By utilizing CrA–biotin and fluorescein–biotin in the following study [163], they labeled Clostridium perfringens enterotoxin (cCPE) for both xenon MRI and fluorescence detection. cCPE has a high affinity tospecific members of the claudin (Cldns) family, and can serve as the targeting unit for a claudin-sensitive cancer biosensor. HyperCEST images were acquired using a 30 μT, 20 s saturation pulse. HyperCEST imaging correctly localized the cells expressing Cldn4 in the inner compartment of the phantom, as demonstrated by the high HyperCEST effect (Figure 5). The results of a 59% HyperCEST for Cldn4-expressing HEK cells and 11% HyperCEST for non-transfected HEK cells were obtained. In subsequent work [161], CrA was functionalized with a fluorophore and bicyclo[6.1.0]nonyne to target cell-surface glycans.

The successful demonstration of this concept within living cell structures offers a perspective for translating the approach into *in vivo* studies. Zeng et al. [129] have recently proposed a protocol for detecting substrates in living cells using targeted molecular probes through hyperpolarized ^129^Xe MRI. The authors present an optimized protocol utilizing a cryptophane-based probe sensitive to biothiols and demonstrated effective detection and imaging of substrates in human lung cancer A549 cells. However, there is a need for further improvement to achieve accelerated CEST build-up that can effectively overcome the intrinsic loss of hyperpolarization under *in vivo* conditions. In a recent study [91], liposomes with a HyperCEST-active lipopeptide to enhance the efficiency of a CrA-ma host with medium xenon exchange kinetics in an aqueous environment were presented. The use of liposomes with lipopeptide-anchored CrA as the Xe host eliminates the necessity for significantly increasing B_1_ to align with the accelerated exchange kinetics. This enhancement is notable because liposomes create an environment that facilitates a faster turnover of the moderately hydrophobic noble gas by the hydrophobic Xe@CrA host.

Over the past decade, efforts to improve the sensitivity of CrA detection have centered around incorporating multiple ^129^Xe hosts per targeting unit or utilizing hosts with a larger loading capacity. This was first demonstrated in 2006 [164] through attaching a biotin moiety to a polyamidoamine dendrimer, which was able to encapsulate two CrA units when bound to avidin. This increased the SNR by a factor of eight relative to the original CrA biosensor. In the following years [165], attaching multiple CrA units was proposed by using bacteriophage MS2 with further improvements using bacteriophage M13 [166]. Jeong et al. [165] have developed precise, selective, and highly sensitive ^129^Xe nanoscale biosensors by utilizing a spherical MS2 viral capsid, CrA, and DNA aptamers. These biosensors exhibited robust binding specificity to targeted lymphoma cells. The hyperpolarized ^129^Xe NMR signal contrast and HyperCEST ^129^Xe MRI image contrast demonstrated their potential as highly sensitive hyperpolarized ^129^Xe biosensors for future applications *in vivo* cancer detection.

Despite recent advancements, there remains a need for significant optimization in the design of cryptophane-based biosensors. Cryptophane cages are regarded as the benchmark for xenon biosensor applications due to their exceptional ^129^Xe binding constant. However, their commercial unavailability, complex synthesis involving multiple steps, and low yields pose some challenges. While cryptophane-A is difficult to synthesize, the biocompatible and readily available cucurbit[6]uril [104,167,168] has been recognized as a promising alternative. In the next subsection, we delve into cucurbit[6]uril’s biosensing properties.

### 3.2. Cucurbit[6]uril

#### 3.2.1. Cucurbit[6]uril Structure and Properties

Cucurbit[6]uril (CB6) belongs to the family of macrocyclic compounds known as cucurbit[n]urils, which are characterized by the presence of six glycoluril units [169,170]. Its cylindrical shape, featuring a hydrophobic cavity (outer diameter ~5.5 Å), enables the binding of lipophilic molecules. The entrances to the cavity are surrounded by polar ureido carbonyl groups [171,172,173,174,175]. Since the atomic radius of the xenon atom is 1.31 Å, CB6 is a suitable host for HP ^129^Xe complexation.

The binding affinity between Xe and CB6 was measured for the first time by Haouaj et al. in 2001 by ^1^H NMR in an aqueous Na_2_SO_4_ solution and was found to be ~200 M^−1^ [176,177]. However, the low solubility of CB6 in pure deionized water (<10^−5^ M^−1^) [178] creates a challenge for the investigation of kinetic binding between CB6@^129^Xe and *T*_1_ of HP ^129^Xe inside the CB6. Kim et al. [179] resolved this issue by synthesizing a CB6 derivative with cavity dimensions identical to those of the unaltered CB6. The derivative exhibited a significantly higher solubility of 2 × 10^−1^ M in pure water. Using isothermal titration calorimetry, they measured the binding constant to be (3.4 ± 0.1) × 10^3^ M^−1^, with an enthalpy of 2.3 ± 0.3 kcal mol^−1^ and entropy $T∆S^{{}^{\circ}}=2.4\pm 0.3$ kcal mol^−1^. This indicates that the complexation of HP ^129^Xe with CB6 is caused by both enthalpy and entropy and might happen due to the van der Waals interaction between HP ^129^Xe and the inner wall of CB6 and the removal of water molecules from the cavities. Additionally, the HP ^129^Xe NMR spectrum was acquired with CB6 in pure water. The free HP ^129^Xe signal was observed at 190 ppm, while the signal from HP ^129^Xe encapsulated in CB6 had a chemical shift of 97 ppm relative to the gas phase. Therefore, the substantial difference in the chemical shift between the two pools of HP ^129^Xe has proven the effectiveness of the utilization of CB6 as a molecular cage.

The Kim et al. investigation of CB6 as a molecular cage for HP ^129^Xe was followed by Wang et al. [98] who proposed CB6 as a HyperCEST molecular agent. They investigated CB6 in two physiologically relevant solutions: a phosphate buffer solution (PBS) and human plasma. The PBS served perfectly for the investigation due to the increased solubility of CB6 in water in the presence of monovalent cations. The NMR spectrum of 5 mM CB6 in PBS demonstrated a 72 ppm chemical shift for HP ^129^Xe bound to CB6 (Xe@CB6); interestingly, the chemical shift of Xe@CB6 in human plasma was found to be the same. The suitability of CB6 for HyperCEST was tested with the application of Dsnob-shaped saturation pulses over the 85–210 ppm chemical shift range with a step of 5 ppm in 0.8 µM CB6 in PBS. The observed HyperCEST effect was around 50%. Due to the nature of the CB6 shape, CB6 can be potentially blocked by other molecules in the biological media. Accounting for this, Kim et al. [179] performed a HyperCEST experiment on CB6 in human plasma in the presence of putrescine, which is the most abundant polyamine in biological fluids with a high affinity for CB6. They compared the HyperCEST effects of 1 µM CB6 in PBS, human plasma, and human plasma with 10 µM putrescine, which was found to be 50%, 30%, and 15%, respectively. Additionally, they determined the association constant of HP ^129^Xe and CB6 as 490 M^−1^ at 300 K in PBS with NMR exchange spectroscopy and calculated the depolarization rates during on- and off-resonance saturation to be τ_on_ = 24.6 ± 1.2 s and τ_off_ = 58.5 ± 3.7 s.

Hane et al. [99] switched to whole bovine blood as the media for HyperCEST testing. They acquired an NMR spectrum of 2.5 mM CB6 dissolved in a mixture of blood and PBS with a 3.0 T clinical MRI scanner. There were three peaks observed: 192.4 ppm peaks which correspond to the HP ^129^Xe gas peak, a 124.3 ppm peak of Xe in CB6, and a 219.4 ppm peak from Xe bound to red blood cells (RBCs). The HyperCEST depletion spectrum in blood was acquired after the application of sixteen 6 ms 3-lobe sinc saturation pulses with a 3 ms pulse interval. A HyperCEST effect of 74% was observed in a 2.5 mM CB6 solution; the decreasing of CB6 concentration resulted in achieving a detectability limit of 250 µM CB6 with 14% depletion. Moreover, Hane et al. [99] acquired MR images of two syringes filled with CB6 dissolved in blood and with pure blood without the addition of CB6 to produce the saturation map.

The first exploration of kinetics in a Xe@CB6 system was performed by Kunth et al. [152] in 2015. The data acquisition employing a qHyperCEST approach was conducted with varying saturation pulse strengths in pure water. In particular, continuous-wave saturation with a specific strength was applied for a certain duration. This allowed the assessment of the HP ^129^Xe exchange kinetic and binding paraments, such as the ratio of bound to free Xe, Xe exchange rate, Xe association constant, and Xe@host occupancy. The concentration of CB6 in water was 3.4 μM, whereas the Xe concentration was determined based on the Xe pressure and Ostwald solubility. The results revealed the high occupancy (49%) of CB6 by HP ^129^Xe; the exchange rate was calculated to be 2100 ± 300 s^−1^, while the binding constant was found to be 2500 ± 400 M^−1^. Additionally, the study demonstrated that an increase in the saturation pulse strength significantly boosts the HyperCEST effect for CB6.

Korchak et al. [180] utilized another approach for the assessment of the exchange kinetics: they varied the pressure of the HP ^129^Xe (and hence its concentration) in the 4.5 mM CB6 in PBS solution. Two primary mechanisms were taken into consideration: (1) HP ^129^Xe spontaneously leaves the CB6 complex or is replaced by a guest molecule and (2) Xe is replaced by another Xe without the generation of Xe-free CB6. They used a modified approach which included a strong rectangular RF pulse tuned for both (1) the inversion of free HP ^129^Xe magnetization and (2) rotation of the off-resonant Xe-CB6 magnetization for the full cycle around the applied effective field back to the original position. They also employed varying time intervals between the preparation RF pulses and signal readout to calculate the depletion rates of free HP ^129^Xe (2575 ± 76 s^−1^) and Xe@CB6 (2449 ± 43 s^−1^). The exchange rates were estimated as 1131 ± 11 s^−1^ for the first mechanism and 108,600 ± 8700 M^−1^ s^−1^ for the second mechanism; the affinity constant was 289 ± 8 M^−1^.

Later, Mitschang et al. [181] combined two approaches used previously in one comprehensive study to determine the exchange parameters in absolute terms. They modulated both a concentration of HP ^129^Xe within the solution and an amplitude of the applied saturation RF pulses. Additionally, pulses were applied at different Xe nutation frequencies with different durations. All experiments were conducted with a 0.0165 mM CB6 solution in PBS. The built model included both exchange mechanisms mentioned earlier. Using the variation of two parameters (Xe pressure and RF amplitude), they evaluated the rate constants for both mechanisms: the affinity constant and total host molecule concentration. All results were in good agreement with the parameters measured by previous studies using the variation of only one parameter (Xe concentration within the solution).

#### 3.2.2. HyperCEST Imaging Using CB6

The successful detection of CB6 HyperCEST *in vitro* initiated translation to *in vivo* studies. The pioneering research by Hane et al. [100] established the foundation for the novel applications of HyperCEST biosensors within *in vivo* studies, paving the way for potential translation into clinical applications. Hane et al. were the first to perform studies with CB6 *in vivo* using the clinical 3.0 T MR scanner. They injected 3 mL of a 10 mM solution of CB6 in PBS into a living rat and acquired MR spectra of the rat’s abdomen and head separately. They observed three peaks at 184 ppm, 192.5 ppm, and 207 ppm with respect to the gas frequency at 0 ppm in the rat abdomen which was assigned to the fat, lung parenchyma, and blood, respectively. The spectrum for the head revealed four Xe peaks at 185 ppm, 191 ppm, 193 ppm, and 205 ppm, which were assigned to the muscle, white matter, grey matter, and red blood cells, respectively. Sixteen 20 ms 3-lobe sinc depolarization pulses with 3 ms intervals were used for HyperCEST depletion. A HyperCEST effect of 22% was observed in the rat abdomen and 55% depletion was observed in the rat brain. HyperCEST saturation maps were calculated from the HP ^129^Xe MR images acquired following off-resonance and on-resonance saturation (Figure 6). While Hane et al. detected CB6 in the rat brain, it is essential to investigate CB6’s ability to cross the blood–brain barrier (BBB) before considering its application as a molecular biosensor in living organisms. Thus, Newman et al. [182] confirmed the crossing of the BBB by CB6 using ^1^H and ^13^C NMR within an artificial brain sphingomyelin-based BBB.

McHugh et al. [102] conducted another *in vivo* study with CB6 in mice using a 9.4 T small animal MR scanner. They studied the effect of the number of breaths on HP ^129^Xe on the HyperCEST intensity. A 0.1 mL volume of 20 mM CB6 in PBS was injected into the mice. Sixteen 3-lobe sinc pulses with a 20 ms duration and 400° flip angle were applied. Additionally, a second injection of 0.1 mL CB6 solution was made 1 h after the first injection. They observed a decrease in the HyperCEST effect from 15% to 5% between one breath and four breaths of Xe after the first CB6 injection and a decrease from 40% to 20% between the same number of breaths after the second CB6 injection. HP ^129^Xe on-res and off-res images were acquired following one injection of CB6 for the different number of breaths. The corresponding HyperCEST maps were calculated (Figure 7).

The latest *in vivo* study was performed by Kern et al. [104] in 2023 in rats using a 2.89 T MRI scanner. They obtained chemical shift (CSI) HyperCEST maps of HP ^129^Xe in lipids, aqueous tissues, and RBCs (Figure 8) after the injection of 5 mM CB6 solution in saline (10 mL/kg body weight). They utilized 10 rectangular 50 ms long saturation pulses (~24 μT) with 1 ms gaps.

#### 3.2.3. Exploring Potential Functionalization of CB6

While the CB6 molecular cage proved highly effective in HyperCEST, the addition of an affinity tag is necessary to transform it into a molecular biosensor. However, this poses a significant challenge as CB6 is symmetrical and very stable. The only attempt at CB6 covalent functionalization was performed by Prete et al. [183] in 2018. They achieved a monofunctional CB6 derivative by attaching a benzene to it. The modified CB6 demonstrated a 71.5% depletion for a 10 mM solution of CB6 derivative in DI water using a sinusoidal saturation pulse.

All other endeavors to transform CB6 into a molecular biosensor involved creating rotaxanes, where CB6 is mechanically obstructed by a “dumbbell” stopper. This stopper cleaves CB6 under specific conditions, facilitating HyperCEST. The conditions of cleavage or activation may be varied with the different stoppers whose decomposition can be controlled. The first study which utilized a similar rotaxane approach was conducted by Wang et al. [184] in 2016. They designed two-faced guests which effectively blocked the CB6 cavity from Xe. One side of these guests contained a butylamine tail which has a high affinity for CB6, and the other side consisted of *p*-benzenesulfonamide, which is known to have an affinity to human carbonic anhydrase II. The effectiveness of the designed molecular biosensor was proven by first conducting a HyperCEST experiment with a 1 µM solution of CB6 in PBS resulting in 50% depletion. Next, 4 µM of the two-faced guest was added to the solution which blocked the CB6 cavity. Consequently, the HyperCEST depletion decreased to 15%. In order to free CB6, 4 µM of human carbonic anhydrase was added, which restored the HyperCEST effect of 30%. The performance of the CB6 biosensor was also investigated in two kinds of BL21(DE3) *E. coli* cells: non-transformed and expressing recombinant human carbonic anhydrase II. Concentrations of 4 µM and 16 µM of the two-faced guest and CB6, respectively, were added to both cell types. The observed HyperCEST effect was 20% in the control cells and 45% in the cells with a high concentration of human carbonic anhydrase II.

The first actual rotaxane with CB6 was proposed by Finbloom et al. [185] in 2016. They synthesized rotaxane with pyrene-functionalized 2-azidoethylamine and an adamantly-ester-functionalized propargylamine as stoppers with β-cyclodextrin caps for solubility improvement. It also had a labile ester group which can be hydrolyzed to leave CB6. Impressive results were observed with no HyperCEST detected in the presence of rotaxane and almost 100% depletion after adding 10 equivalents of LiOH to release the CB6. This work was expanded by Slack et al. [186] in 2017 through the creation of a rotaxane that can be activated by matrix metalloprotease 2 (MMP-2). The rotaxane had pyrene and (5,6)-carboxytetramethylrhodamine stoppers and the PLG-LAG recognition sequence of MMP-2. A 5 µM concentration of rotaxane in dH_2_O presented no depletion without MMP-2 and a 25% HyperCEST effect after the addition of 5 nM MMP-2. Another rotaxane designed by Klass et al. [187] was capable of detecting elevated H_2_O_2_ levels. It was capped with a maleimide functional group and was sensitive and selective to the H_2_O_2_ aryl boronic acid group, while the rotaxane axle consisted of a p-xylenedi-amine moiety. Unfortunately, only 25–28% HyperCEST depletion was observed upon release of CB6 in a 25 µM rotaxane solution in PBS with the addition of 50 µM H_2_O_2_. The study also detected the endogenously produced H_2_O_2_ in HEK 293 T cells which were exposed to tumor necrosis factor alpha. A 10 µM concentration of rotaxane present in the cells treated with tumor necrosis factor alpha significantly broadened the Xe-CB6 signal depletion over time while untreated cells did not show any HyperCEST depletion.

Finally, Dopfert et al. [188] used ultrafast HyperCEST with CB6 for dynamic monitoring of an enzymatic reaction course and to calculate its rate and specific activity. This was possible due to the ability of CB6 to reversibly bind the product of the reaction in addition to HP ^129^Xe, which significantly decreased the HyperCEST effect of CB6. The reduction in the HyperCEST effect was proportional to the reaction product concentration. The acquisition of ultrafast HyperCEST spectra over time after the beginning of the reaction allows us to monitor the formation of the reaction product and calculate the kinetic properties. The reaction studies by Dopfert et al. investigated the conversion of lysine to cadaverine induced by the enzyme lysine decarboxylase due to the ability of CB6 to reversibly bind cadaverine.

All the abovementioned studies utilized different saturation pulses for the HyperCEST evaluation, and hence, their results cannot be compared quantitively. Indeed, the type and strength of the depolarization pulses play a crucial role in CB6 performance as a molecular cage for HyperCEST. This was recently proven by Grynko et al., who compared the performance of four different depolarization pulses: sinusoidal, block, 3-lobe sinc, and hyperbolic secant (Figure 9) [103]. They also tested a variation of the saturation pulse FAs and determined that the highest achievable FA provides the best results. The application of a 3-lobe sinc pulse with a 1530° FA demonstrated the highest performance for CB6 HyperCEST. In addition, the CB6 detectability limit was investigated in whole bovine blood and was determined to be 100 µM using a 3.0 T MRI scanner. The HyperCEST effect in red blood cells was observed for the first time in this study.

While promising developments have been observed with CB6 and CrA, it is essential to note that each exhibits distinct characteristics. However, the choice between these xenon host molecules depends on the specific application and the desired attributes for a given biosensing scenario. In the realm of ^129^Xe MRI, there is a notable focus on harnessing ^129^Xe@host chemistry for biosensing applications. This approach aims to utilize genetically encoded ^129^Xe biosensors, including structures such as gas vesicles and monomeric proteins equipped with hydrophobic cavities, allowing for transient interactions with xenon.

### 3.3. Gas Vesicles

There has long been an interest in developing biologically engineered, genetically encoded, optical receptors for usage in HyperCEST MRI. Synthetic biosensor constructs have proven difficult to deliver to cells, have required extremely high concentrations, and remain challenging to connect to biological processes [95,100]. Developing HyperCEST biosensors that are genetically engineered and encoded by genes would be ideal for their potential to link to biological processes [189]. Gas vesicles (GVs) are naturally occurring HyperCEST agents that are produced by a range of buoyant microorganisms, such as algae and bacteria. Gaseous vesicles (GVs) are approximately 50–500 nm in size, with a 2 nm thick protein shell [190,191]. GVs are traditionally used by these microorganisms to regulate their flotation and buoyancy when searching for nutrients. Biologically, GVs are made of at least eight protein, most of which is occupied by GvpA, which is a highly conserved gas vesicle structure protein [192]. GvpB and GvpC gas vesicle proteins are also involved in supporting the external scaffolding of the vesicle. GVs have been isolated from *Anabaena flos-aquae*, *Halobacterium NRC-1*, and *Escherica coli* [97,193]. GVs have long been theorized to be able to encase hyperpolarized agents for usage in conjunction with HyperCEST MRI. GVs are a well-established ultrasound agent, and as of recently, they have been gaining attention for their potential as an MRI intravascular contrast agent. GVs are an ideal candidate for HyperCEST MRI as dissolved xenon gas can diffuse into GVs, where the xenon is then able to rapidly exchange between the solution and GV. GVs are gas-binding protein nanostructures. GVs consist of a protein coat that encapsulates a hollow interior filled with gas. GVs are composed of a heavy, dense gas-filled core, surrounded by a lipid shell [190]. GVs were first reported in medical imaging for their usage as an ultrasound contrast agent [194,195,196,197]. GVs work in ultrasound by scattering sound waves, which can be translated to visual resolution with ultrasound. Interestingly, GVs have also been adapted acoustically to allow for selective acoustic modulation of cells using ultrasound [198].

Shapiro et al. [97] first reported the usage of GVs in HyperCEST xenon MRI in 2014. They demonstrated that using GVs, they could provide a stable environment for xenon (and other noble gases), as well as increase the resolution by 100–10,000-fold when compared to similar ^1^H MRI constructs. They also showed that GVs were effective at CEST with xenon, which allowed for the GVs to be visualized within the picomolar range. GV contents can reflect the gases that are dissolved within the surrounding liquid. GVs can clasp hyperpolarized ^129^Xe, which is then detectable using HyperCEST pulse sequences. Farhadi et al. [193] have also expanded the scope of GVs to include Mega CVs, which are 32 times smaller than conventional GVs by volume, yet still able to produce the same xenon MRI contrast. It was also shown that the rapid rate of xenon exchange in and out of GVs may cause a broadening of the peak saturation. Farhadi et al. showed that the HyperCEST effect when combined with GVs follows a GV concentration-dependent exponential relationship.

The thin protein shell of GVs allows for gases to freely diffuse in and out of the interior [199]. Additionally, water is prevented from condensing into liquid form within the core. GVs have the advantage of having a very high rate of xenon diffusion when in the gaseous phase compared to other HyperCEST biosensors, such as perfluorocarbon nanoemulsions [95]. GVs also allow for a much larger chemical shift between the solvent and gaseous phase peak within the GV core [193]. GVs are also of the optimal size for HyperCEST efficiency. GVs can be customized depending on the biological process of interest, and their shell chemistry can be modified by adding or subtracting any combination of proteins, lipids, polysaccharides, or polymers. All these molecules have different xenon solubilities, which can change the xenon exchange rate in and out of the GV. GVs have a unique ability to collapse when under specific amounts of pressure, which allows for background-subtracting and multiplexed imaging [193]. This allows for straightforward experimental controls, as GVs do not produce contrast when collapsed.

Other biologically engineered HyperCEST adjuncts have been developed, such as the green fluorescence protein (GFP), which have yielded HyperCEST sensitivities within the micromolar range (mM), but GVs have been shown to yield sensitivities in the picomolar range (pM) [200]. GVs were one of the first demonstrations of the application of biologically engineered molecules due to their potential for combining with HyperCEST MRI to visualize cellular mechanisms at a molecular level.

GVs have significant potential for allowing researchers and clinicians to visualize molecular processes at the cellular level. This entails allowing for the visualization of cell signaling pathways, immune system modulation, and cancer cell metastasis, amongst many other processes that involve cell migration, cellular growth, or gene expression. The first *in vitro* demonstration of this was also reported by Shapiro et al. [97] and featured the visualization of quantitative gene expression in *Escherichia coli* (*E. coli*) and the visualization of the presence of the HER2 protein in breast cancer cells. Shapiro et al. also used pharmacokinetic model predictors to determine whether *in vivo* GV HyperCEST xenon MRI would be achievable. Their results supported the feasibility of *in vivo* GV HyperCEST imaging.

GVs have demonstrated considerable potential in preclinical studies, yet the imminent challenge lies ahead. The feasibility of GVs needs validation in imaging scenarios employing model organisms and subsequent clinical trials involving human participants [201]. Currently, engineering GVs requires specialized cell culture techniques with cyanobacteria or haloarchaea, which is limiting the genetic engineering of GVs [193]. Xenon delivery will also have to be analyzed, including the mode of delivery, timing, route, and kinetics. With continuing advances in xenon MRI technology, it is reasonable to expect that combining biological adjuncts with HyperCEST xenon MRI will one day make the imaging of biological and molecular processes possible.

### 3.4. Recently Discovered Novel HyperCEST Agents

Since conventional HyperCEST-active macrocycles are either difficult to functionalize to create a full-fledged biosensor (e.g., CB6) or difficult to synthesize with a high yield (e.g., CrA), the search for new and better HP ^129^Xe molecular imaging agents is ongoing. As was mentioned previously, there are four main criteria that any potential candidate molecule should satisfy in order to be considered a successful candidate for future HyperCEST biosensor development for practical applications: (a) being water soluble; (b) being easy to functionalize; (c) possesses a strong HyperCEST effect *in vitro* in aqueous solutions; and (d) has distinct chemical shifts of HyperCEST peaks from other dissolved phase ^129^Xe resonances.

Recently, a different class of compounds—metal–organic capsules—were proposed as promising HyperCEST active agents [96,202]. Specifically, capsules with the general formula [M^II^_4_L_6_]^4−^, where L^2−^ is 4,4′-bis [(2-pyridinyl-methylene)amino]-[1,1′-biphenyl]-2,2′-disulfonate, possess a high affinity to ^129^Xe in aqueous solutions. These metal–organic capsules can be easily synthesized in one step with high yields via self-assembly and due to the presence of commonly utilized ferromagnetic metals such as Co and Fe; they also demonstrate a distinct chemical shift of ^129^Xe@ [M^II^_4_L_6_]^4−^ resonance from the gas phase and dissolved phase signal [96,202,203]. Jayapaul et al. demonstrated the existence of three different HyperCEST ^129^Xe@ [Fe_4_L_6_]^4−^ depletion peaks at +13, +24, and +30 ppm with respect to the dissolved phase ^129^Xe resonance. These multiple peaks’ origin was explained by ^129^Xe interactions with different diastereomers of the [Fe_4_L_6_]^4−^ capsule [113]. The HyperCEST depletion decreased gradually from ~80% for the +13 ppm resonance down to ~ 20% for the +30 ppm peak for a 100 μM concentration of [Fe_4_L_6_]^4−^ at 25 °C. On the contrary, the [Co_4_L_6_]^4−^ metal–organic capsules demonstrated a single HyperCEST depletion peak at −89 ppm with respect to a dissolved-phase ^129^Xe resonance in H_2_O [96] due to the high-spin Co^2^ atoms. The intensity of the HyperCEST effect was over 85%. Interestingly, further substitution of Co atoms for Fe atoms allows with creation [Co_4-n_Fe_n_L_6_]^4−^ allows fine tuning of the cage resonance frequency which may be useful for further practical applications *in vivo* [203]. All ^129^Xe@ [M^II^_4_L_6_]^4−^ HyperCEST peaks demonstrated a strong linear temperature dependence with ^129^Xe@ [Co_4_L_6_]^4−^ being the most temperature-sensitive and ^129^Xe@ [Fe_4_L_6_]^4−^ being the least [203].

Overall, [M^II^_4_L_6_]^4−^ has been demonstrated to be quite a promising framework for further development of HyperCEST molecular imaging biosensors with finely tunable chemical shifts. One of the major advantages of these metal–organic frameworks and, specifically, [Co_4_L_6_]^4−^ is their HyperCEST insensitivity to the presence of biological fluids [96]. Non-selective binding of biomolecules is one of the major limitations of the CB6 supramolecular cage, which makes [M^II^_4_L_6_]^4−^ one of the best candidates for future implementation in *in vivo* molecular HyperCEST imaging. Despite the high HyperCEST performance and extremely low non-selective binding of biomolecules, these promising ^129^Xe molecular agents should undergo extensive biocompatibility and toxicology studies due to the presence of heavy metals in them. Moreover, the presence of the paramagnetic cations may also result in potential relaxation effects that will affect the overall HyperCEST performance of the [M^II^_4_L_6_]^4−^ biosensors. Thus, future relaxometry studies are needed to evaluate these potential relaxation effects at different magnetic field strengths.

Another type of macrocycle that has recently shown potential for HyperCEST molecular imaging applications is pillar[5]arenes (P[5]A) [93,101]. Water-soluble pillar[5]arenes synthesized by appending carboxylate groups at both rims with different counterions (Na^+^, Br^−^, and NH_4_^+^) demonstrated a moderate magnetization transfer (MT) performance of ~47% which still significantly outperformed the CrA reference sample that demonstrated only an ~29% CEST performance [93]. A more recent study evaluated the HyperCEST performance of decacationic P[5]A with ten cationic imidazolium groups attached to both rims [101]. It was found that decacationic P[5]A possesses a relatively weak HyperCEST effect of ~23% at −77 ppm with respect to the dissolved phase ^129^Xe signal. Despite the relatively weak HyperCEST and MT performance of P[5]As due, presumably, to fast exchange between the cage and the dissolved ^129^Xe pool, P[5]As are easy to synthesize with reasonable yields and are potentially easy to functionalize compared to CB6 and CryA cages. Moreover, P[5]As engage in a 1:1 interaction with HP ^129^Xe which minimizes potential relaxation effects on HyperCEST performance. However, similar to other HyperCEST agents, the biocompatibility of water-soluble P[5]As is still unknown and remains a subject for future studies.

A new take on potential HyperCEST agents has been proposed recently with the introduction of microbubbles as a potential HyperCEST contrast [95]. These lipid-shelled perfluorocarbon microbubbles rely on breaking the 1:1 interaction between the dissolved ^129^Xe pool and the contrast agent, which should facilitate RF depolarization of ^129^Xe nuclei. Since microbubbles contain HP ^129^Xe in a gas phase, the depolarization pulses are applied close to the gas phase (at −194.6 ppm with respect to the dissolved ^129^Xe phase). The HyperCEST performance of microbubbles depended on their size and varied from ~26% for microbubbles with an average diameter of less than 1 µm up to ~94% for microbubbles with an average diameter of 2 µm. It should be noted that the microbubbles’ detectability limit was ~fM at 11.8 T. Despite the overall outstanding performance of the microbubbles in terms of the HyperCEST strength and detectability limit, the main challenge that may cast a shadow on the future utilization of microbubbles as HyperCEST molecular imaging agents is a potential challenge in their functionalization through molecular targeting. On the other hand, lipid-scaled microbubbles have been already approved by the FDA, unlike any other HyperCEST-active molecular imaging agents.

Finally, a very recent study demonstrated a novel type of water-soluble supramolecular agent with outstanding HyperCEST performance—R3-Noria-Methanesulfonate (R3-Noria-MeSO_3_H) [204]. The original Noria is a waterwheel-shaped molecule with six small peripheral cavities and one large center hydrophobic cavity originally designed to selectively bind xenon [205]. Despite the water insolubility of conventional Noria and its isomer R3, the designed R3-Noria-MeSO_3_H is highly water soluble and demonstrated a HyperCEST effect of up to ~85% in aqueous solutions using a 3.0 T clinical MRI scanner. Besides high HyperCEST contrast, R3-Noria-MeSO_3_H has demonstrated a superior *T*_2_*** contrast that opens a new possibility to utilize it for HP ^129^Xe molecular MR imaging (Figure 10).

Unlike any other large supramolecular HyperCEST contrast agent, R3-Noria-MeSO_3_H demonstrated quite interesting molecular aggregation dynamics by forming molecular clusters with a mean size of up to ~200 nm in aqueous solutions. The authors suggested the possibility of R3-Noria-MeSO_3_H–R3-Noria-MeSO_3_H direct interactions in deionized water as well as cation-mediated aggregation in the presence of the dissolved salts. These aggregation dynamics and the strong downfield shift of the HyperCEST peak (+90 ppm with respect to the dissolved phase ^129^Xe resonance) suggested the predominant binding of HP ^129^Xe in the outer hydrophobic pockets of R3-Noria-MeSO_3_H. Interestingly, the aggregation of R3-Noria-MeSO_3_H may potentially result in binding multiple ^129^Xe atoms to one molecular cluster which may result in a high effectiveness for this molecular imaging agent. On the other hand, the molecular aggregation of R3-Noria-MeSO_3_H should be carefully studied prior to further transition of this HyperCEST agent to preclinical molecular imaging to ensure the safety of intravenous injections of R3-Noria-MeSO_3_H solutions. In addition, the identification of the HP ^129^Xe binding sites on R3-Noria-MeSO_3_H aggregates as well as the relaxation mechanisms of the bounded HP ^129^Xe should become the subjects of future studies. The detectability limit of R3-Noria-MeSO_3_H was found to be similar to that of CB6 at the same magnetic field strength and equal to 50 μM. This detectability level may be further improved after dedicated pulse sequence optimization.

## 4. *In Vivo* HyperCEST Imaging

Although the HyperCEST contrast mechanism holds potential for molecular imaging outside the living organism (*ex vivo*), the application of HyperCEST within *in vivo* trials is a relatively recent advancement [100,102,104]. Only a few experiments have been conducted involving *in vivo* HyperCEST imaging, and while the results are promising, there is still much needed exploration of the method of signal acquisition and different contrast molecules before clinical trials will be possible. *In vivo* HyperCEST imaging thus far has only been attempted by three research groups [100,102,104]. All three trials were performed on rats and mice using CB6 as a contrast molecule. All experiments were performed in compliance with the ethical standards and regulations applicable in the respective regions. Given that the technical details and methodologies for all aforementioned studies were discussed in the previous section, we will now explore the perspectives these studies offer in facilitating the transition of HyperCEST imaging into clinical settings.

Hane et al. [100] were the pioneers in utilizing CB6 as a contrast agent for *in vivo* HyperCEST imaging. The injection of CB6 into the tail vein and mechanical ventilation of a xenon gas mix (80% xenon. 20% oxygen), followed by a 30 min biodistribution period of CB6 prior to HP ^129^Xe MRS and HyperCEST image acquisition was performed. HyperCEST saturation maps were superimposed on a ^1^H MR image to depict the positioning of the CB6 cage contrast agent. The study faced limitations, including difficulties in attaining the intended pulse lengths, challenges in measuring the interactions between CB6 and ^129^Xe@CB6 in specific tissues, and suboptimal optimization of the scanner parameters for depletion spectra. A 3.0 T clinical MR scanner was used, which provided both the proof of concept of the HyperCEST imaging in *in vivo* settings utilizing a clinical scanner and constraints inherent to clinical approaches.

Subsequent to Hane et al., another *in vivo* HyperCEST imaging study was performed by McHugh et al. [102] in 2021. The focus of this study was to determine the optimal experimental protocol to achieve consistent *in vivo* HyperCEST contrast. Using a 9.4 T small animal scanner, z-spectra and HyperCEST contrast maps were reconstructed [102]. The z-spectra indicated that maximal contrast came from ^129^Xe saturation on every breath while CB6 was present, and the contrast was reduced as ^129^Xe was saturated every 2, 3, or 4 breaths [102]. The HyperCEST contrast maps of ^129^Xe saturation every breath compared to ^129^Xe saturation every 3 or 4 breaths confirmed the maximal contrast from saturation every breath that was observed in the z-spectra [102]. The ideal conditions allowed for the coordinated control of the ventilation and MR acquisition, as well as repetition of the breathing cycle until a steady-state dissolved-phase ^129^Xe magnetization was reached. Unfortunately, human ^129^Xe images are taken through or after one breath-hold, which would lead to poor, unreliable results using the HyperCEST contrast approach investigated. Additionally, contrast for *in vivo* studies requires large amounts of CB6 for which all of the short-term and long-term effects on humans are unknown. However, research involving the specific properties of CB6 and its potential toxicity was outside the scope of the study.

The most recent results by Kern et al. [104] have shown the feasibility of HyperCEST z-spectroscopy and imaging in spontaneously breathing rats. A study was performed using a 2.89 T MRI with non-functionalized CB6 as a contrast agent to perform chemical shift imaging (CSI) [104]. At least three resonances in the ^129^Xe dissolved phase around 200 ppm were clearly visible and were associated with ^129^Xe in the three phases of RBCs, aqueous tissues (TP), and lipids, whereas gaseous ^129^Xe was not discernible. The HyperCEST CSI indicated that the best contrast was in TP, which is supported by the z-spectra results, followed by the RBCs and the lipids. This study proved the possibility of HyperCEST imaging in a clinically relevant field (2.89 T) using spontaneous breathing with less excessive amounts of CB6 (ca. 5 mM). The reliability of the process, however, is questionable due to the results shown in the report. Almost all results were shown for the 8th rat tested, and it is unclear whether all results were consistent across the other rats. Regardless, the successful contrast images from a spontaneous breathing technique are a much-needed step towards potential clinical trials. The recent findings, therefore, give more hope for *in vivo* and eventually clinical HyperCEST trials. However, they also make clear the current limitations that must be overcome before it can be used for personalized medical care.

However, the potential translation of HyperCEST approaches using CB6-based biosensors to the clinical side will encounter numerous challenges. The HyperCEST contrast magnitude is influenced by the relative concentration of CB6, timing of applying saturation pulses, and exchange kinetics during the breathing cycle. The saturation of ^129^Xe@CB6 has a partial impact on the dissolved-phase pool, primarily because of the relatively slow exchange rate between tissues and blood. Furthermore, CB6 is prone to bind a wide variety of analytes that can hinder the exchange of ^129^Xe.

Although toxicity studies [206,207,208] have been conducted for contrast agents from the cucurbit[n]uril family, specifically for CB7 and CB8, there is currently no available data regarding CB6 in this regard. Furthermore, there is no previous data on the blood half-life of CB6, making it challenging to calculate the dose at a given time after injection. Thus, the successful implementation of CB6 within *in vitro* and pre-clinical settings paves the way for further investigations and assessments for potential clinical translation.

## 5. Discussion

The HP ^129^Xe HyperCEST approach plays a central role within the landscape of molecular MRI. The intrinsic characteristics of ^129^Xe, including its exceptional sensitivity to the molecular environment and high polarizability, pave the way for the development of novel contrast mechanisms that can overcome the limitations of conventional MRI.

The potential dependence of the quality of the HyperCEST MRI image on the dissolved-phase HP ^129^Xe MRI signal also deserves some attention. Although there are no previous studies that evaluated and quantified either the effect of the HP ^129^Xe signal level on the final HyperCEST saturation maps or HyperCEST imaging repeatability (both *in vitro* and *in vivo*), the potential intertwining between HyperCEST molecular imaging performance and dissolved phase ^129^Xe MRI performance can be expected intuitively. Indeed, since HyperCEST imaging relies on the subtraction of two subsequent dissolved-phase HP ^129^Xe images, the image parameters such as the SNR, contrast, and presence/absence of artifacts in the ^129^Xe image will affect the contrast, sensitivity, and SNR of the HyperCEST image. Therefore, future studies dedicated to the overall improvement of the HP ^129^Xe image quality will have to go hand in hand with the future optimization of HyperCEST performance. In general, the maximization of the HP ^129^Xe dissolved-phase signal is beneficial for HyperCEST imaging since it allows for the minimization of the noise-induced subtraction artifacts on the resulting HyperCEST image and improvement in HyperCEST repeatability.

Since the concentration of the dissolved ^129^Xe in aqueous solutions is limited (especially during *in vivo* imaging), it is of utmost importance to increase the polarization level which will result in a linear increase in the HP ^129^Xe image SNR. This can be achieved by enhancing the spin-exchange optical pumping setups through the optimization of the gas handling system. Currently, the highest reported level of ^129^Xe polarization used for HyperCEST experiments was about 56% [204]. Moreover, magnetization loss occurs during each readout pulse for hyperpolarized ^129^Xe. Hence, it is crucial to guarantee the consistent replenishment of xenon magnetization into the setup. Another hardware approach to improving the HP ^129^Xe signal would involve the implementation of a multichannel phased-array RF receiver which is essential to advance HP ^129^Xe HyperCEST molecular imaging. Another advantage of the parallel imaging paradigm implementation is the minimization of the relaxation effects (both *T*_1_ and *T*_2_***) on HyperCEST imaging.

Besides hardware improvement, pulse sequence optimization and development will also directly benefit the HyperCEST performance of all HP ^129^Xe molecular imaging biosensors. The implementation of non-Cartesian k-space trajectories not only allows for increasing the HP ^129^Xe SNR, but it also reduces the *T*_2_*** relaxation effects, although it may cause image blurring due to undersampling of the outer regions of the k-space. It has been previously shown that implementation of the initial 90^0^ depolarization pulse prior to the imaging pulse sequence substantially improved HP ^129^Xe imaging repeatability in human subjects [209]. The same approach may improve overall HyperCEST repeatability, although some pulse sequence modification will be required. In addition, the proper saturation pre-pulse train should be used for maximization of the encapsulated ^129^Xe depolarization, especially in the case of real-life ^129^Xe HyperCEST MRI imaging. While a recent study showed a direct correlation between the depolarization RF pulses’ flip angle (i.e., transmitted RF power) and HyperCEST performance [103], proper pulse shape selection may need to be performed for each HyperCEST imaging agent. Indeed, different supramolecular cages have HyperCEST depletion peaks of different widths and, therefore, different depolarization bandwidths (BWs) are needed to saturate them effectively. In the case of *in vitro* imaging where SAR and B_1_ inhomogeneities are not a primary concern, the high-power 3-lobe sinc pulses have a tendency to outperform the other pulse shapes. However, if *in vivo* imaging is under consideration, the B_1_ inhomogeneity becomes one of the primary issues that can directly decrease the accuracy of the local HyperCEST quantification. Therefore, adiabatic shapes of depolarization pulses may become the better choice since they are insensitive to B_1_ field inhomogeneities. In addition, the duration of the depolarization RF pulse train should be adjusted in order to minimize the SAR and to provide a saturation BW equal or larger to the width of the supramolecular agent’s depletion peak.

Another consideration for *in vivo* HyperCEST imaging that directly affects molecular imaging performance is the breathing protocol. Previously all three general breathing protocols were evaluated: single breath-hold [100], multiple breaths with subsequent breath-hold [102], and continuous breathing [104]. It was demonstrated that the HyperCEST performance becomes maximized once the steady-state for the HP ^129^Xe dissolved phase is achieved. Since the exchange rate between lung tissues and the blood is relatively slow, achieving this relatively steady state takes considerable time and happens after multiple consecutive HP ^129^Xe inhalations. From this perspective, continuous breathing of HP ^129^Xe may be the best approach for achieving the best possible *in vivo* MRI HyperCEST images. Moreover, the implementation of the continuous breathing protocol may allow for the conductance of longer scans, signal averaging, and acquisition of HyperCEST MRI images with higher spatial resolution.

Currently, there are multiple well-established supramolecular HyperCEST-active molecular types that have been studied extensively for molecular imaging purposes. Despite numerous HyperCEST studies conducted in these supramolecular cages, the creation of complete biosensors remains challenging and barely explored. Although there were previous attempts at functionalizing CB6 [183], complete functionalization of this supramolecular cage with an affinity tag for any disease biomarker has been unsuccessful. An alternative approach proposed by Finbloom et al. [185] and Slack et al. [183] involves indirect functionalization of macrocycles via the synthesis of cleavable rotaxanes. Although using this approach a biosensor for matrix metalloprotease 2 has been synthesized, it demonstrated limited HyperCEST performance and required incubation in the presence of matrix metalloprotease 2 for 24 h. On the other hand, despite multiple biosensors being developed based on the CryA supramolecular cage [166], the synthesis procedure for them is extremely complex with a low yield. Currently, there is an urgent need to develop a functionalization strategy for HyperCEST molecular imaging targets that may produce a complete biosensor dedicated to the detection of a specific disease. One of the main requirements for the functionalization of HyperCEST agents is to maintain their HyperCEST performance or minimize the HyperCEST losses. The next leap in HyperCEST molecular imaging is expected once such biosensors can be utilized for *in vitro* or *in vivo* detection of a pathology within a clinical MRI scanner.

Finally, the question of the repeatability of the HyperCEST molecular imaging between different MRI machines remains unanswered. While almost all HyperCEST studies were performed either using NMR spectrometers or animal MRI scanners, the HyperCEST imaging performance is known only for the Philips Achieva 3.0 T MRI scanner (*in vitro* and *in vivo*) [100] and for the Siemens Magnetom Vida 2.89 T MRI scanner (*in vivo*) [104]. Moreover, the studies conducted using these scanners used different pulse sequence parameters and, in the case of animal imaging, different breathing protocols. The HyperCEST repeatability for MRS has been accessed by Grynko et al. [103], whereas repeatability for HyperCEST MRI molecular imaging has not been studied at all. Future studies should focus on assessing HyperCEST performance repeatability for different supramolecular contrast agents within the same MRI scanner as well as assessing the repeatability of the developed HyperCEST imaging protocol between different MRI scanners in order to demonstrate its feasibility for further translation into pre-clinical and clinical studies.

## 6. Future Directions

Despite significant success in discovering and advancing novel HyperCEST-active agents and their *in vitro* applications, there are still many doubts about the feasibility of transitioning to *in vivo* imaging. Translating HyperCEST imaging to the clinical side presents numerous challenges. Firstly, there are no available toxicology studies regarding any of the studied HyperCEST molecular imaging agents or the molecular imaging biosensors based on these agents. Conducting comprehensive toxicology studies is of utmost importance for the field of HP ^129^Xe HyperCEST molecular imaging. Not only will it demonstrate which supramolecular hosts are feasible for future biosensor development and applications, but toxicology studies will also provide a realistic dosing for HP ^129^Xe molecular imaging agent intravenous injections. The knowledge of the allowed dose will provide a great benchmark for HyperCEST sensitivity and detectability limit assessment. Biosensors that display detectability limits in clinically relevant MRI scanners higher than the acceptable dose will be rejected as unfeasible due to the lack of sensitivity. On the other hand, further pulse sequence and hardware optimization studies that aim to successfully detect the HyperCEST from biosensors at concentrations below the recommended injection dosage will provide a unique imaging protocol suitable for *in vivo* imaging.

Furthermore, it is crucial for future studies to understand the biological pathways involved in extracting HyperCEST-active molecular imaging agents from the organism and to determine the time required for their complete extraction. Although three *in vivo* imaging studies have been conducted previously, none of them performed dynamic HyperCEST imaging which could allow for the estimation of biosensor extraction. Future studies should perform a series of dynamic HyperCEST imaging of either supramolecular hosts or biosensor biodistribution and quantify their extraction time and identify the main extraction biological pathways.

## 7. Conclusions

HP ^129^Xe HyperCEST imaging is a promising molecular imaging modality that is currently under extensive development. The total number of studies conducted has increased significantly over the last decade and currently, a multitude of potential HyperCEST molecular imaging agents have been developed and evaluated. In this review, we explored the current advancements and various applications of a broad spectrum of HP ^129^Xe biosensors. It was shown that functionalized and non-functionalized host structures are used extensively for both *in vitro* and *in vivo* studies. Hence, the recent achievements in employing host structures, such as cryptophane-A, cucurbit[6]uril, gas vesicles, and metal-organic complexes, have been pursued. The continuous progress in this area not only showcases the versatility of HyperCEST contrast agents but also underscores its potential impact on advancing molecular imaging capabilities. In recent years, a number of publications have provided substantial evidence supporting the effectiveness of the HP ^129^Xe HyperCEST concept for *in vitro* and pre-clinical studies. Consequently, it is imperative to conduct comprehensive host and biosensor design studies in combination with toxicity studies to facilitate its translation to clinical settings. The successful application of HP ^129^Xe HyperCEST imaging will pave the way for novel approaches for early disease diagnostics, assessment of treatments, drug design, and personalized medicine.

## Figures and Tables

**Figure 1 ijms-25-01939-f001:**
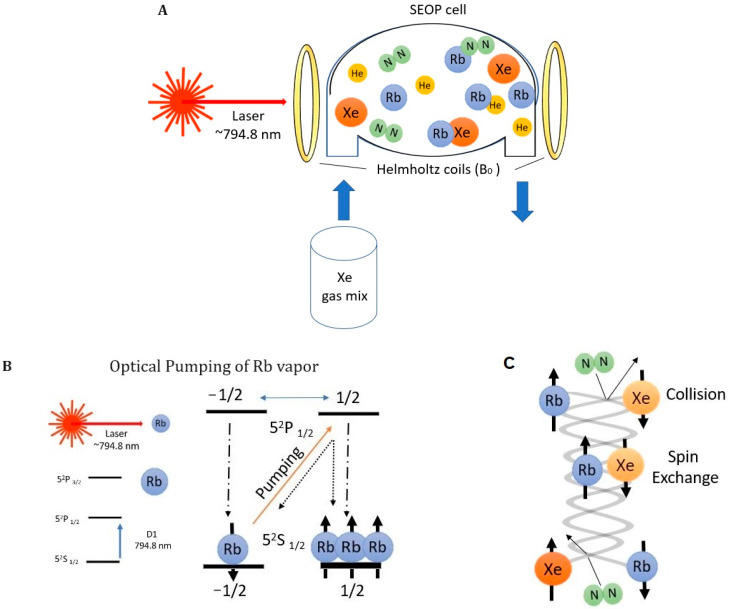
(**A**) Schematic of ^129^Xe spin-exchange optical pumping (SEOP) setup using a high-power Rb laser and optical pumping cell with gas mixture of ^129^Xe, ^4^He, and N_2_; (**B**) the optical pumping of Rb with left-circularly polarized light by D_1_ transition from the ground state 5 ^2^S_1/2_ to in the excited state 5 ^2^P_1/2_; (**C**) the spin-exchange process between the polarized Rb electron spin and the ^129^Xe nuclear spin by the hyperfine interaction in two- and three-body collisions. Adapted from Walker and Happer (1997) [62].

**Figure 2 ijms-25-01939-f002:**
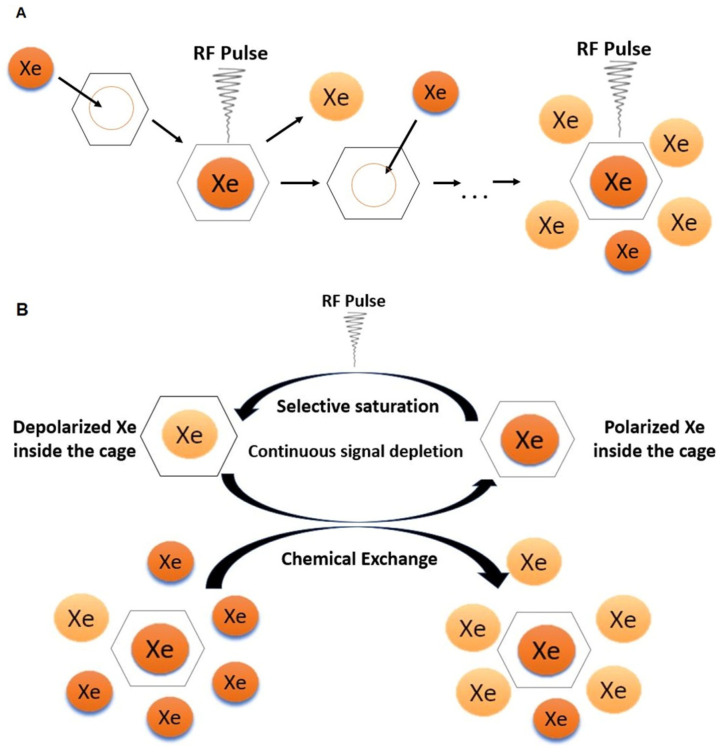
(**A**) Step-by-step diagram on the HyperCEST mechanism: HP ^129^Xe from the dissolved pool goes into the cavity of the host unit; after the reversible enclosure within the host, the selective saturation pulse is applied; depolarized ^129^Xe leaves the cage, and a new HP ^129^Xe atom from pool A takes over the space; the process continues, and thus, in close proximity to the cage, the number of depolarized xenon atoms is increasing. (**B**) The process of chemical exchange saturation transfer between saturated HP ^129^Xe enclosed in a host unit and HP ^129^Xe from the dissolved pool provides continuous signal depletion in the area in close proximity to the host unit, thus causing a greater loss in the net magnetization. Adapted from Schröder et al. (2006) [28].

**Figure 3 ijms-25-01939-f003:**
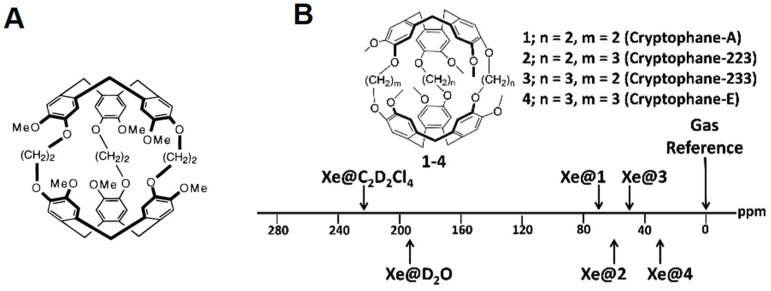
(**A**) Structure of cryptophane-A. Reproduced with permission from Bartik et al. (1998) [142]. Copyright 2023 American Chemical Society. (**B**) Chemical structure and ^129^Xe NMR chemical shifts of cryptophanes with varying alkoxy linker length in 1,1,2,2-tetrachloroethane-d_2_ (1–4). Adapted from Zemerov and Dmochowski (2021) [147] under the CC BY-NC 3.0 DEED License.

**Figure 4 ijms-25-01939-f004:**
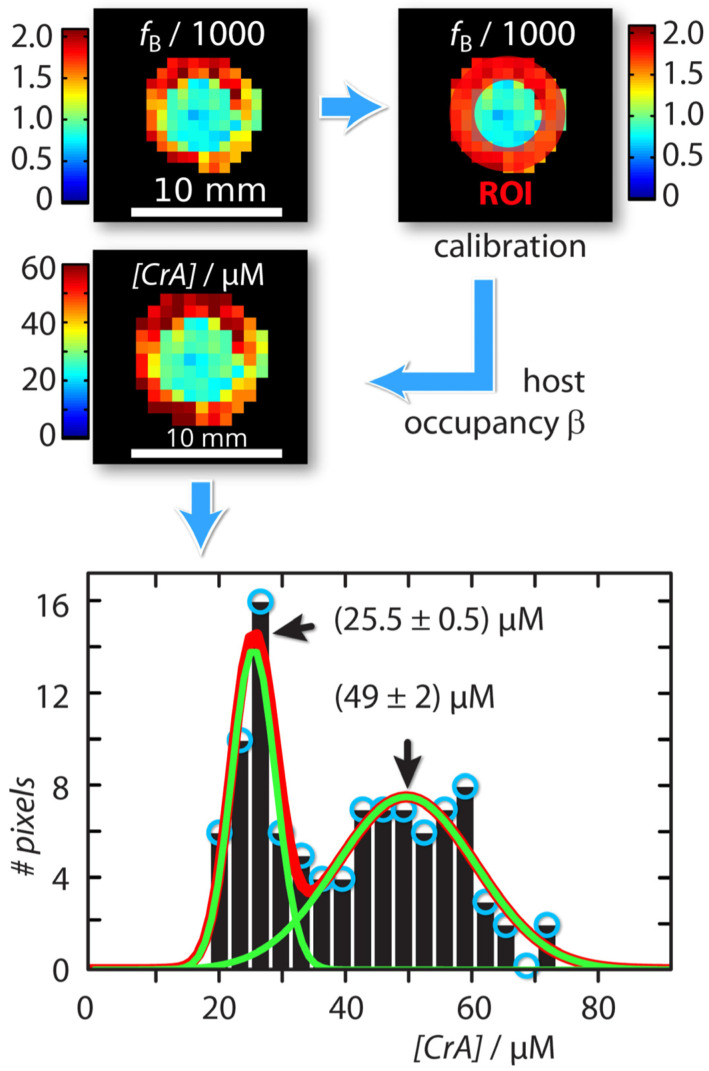
Workflow of absolute concentration mapping starting with the ratio of bound and free Xe,f_β_. At the top left: the host occupancy, β, was calibrated with the HyperCEST response produced by the known reference sample in the outer compartment; the unknown CrA-ma concentration in the inner compartment was then calculated utilizing an internal standard. The Gaussian line fitted histogram of the [CrA-ma] map identified two different concentration populations. Reproduced from Kunth and Schröder (2021) [92] under the CC BY 4.0 DEED License.

**Figure 5 ijms-25-01939-f005:**
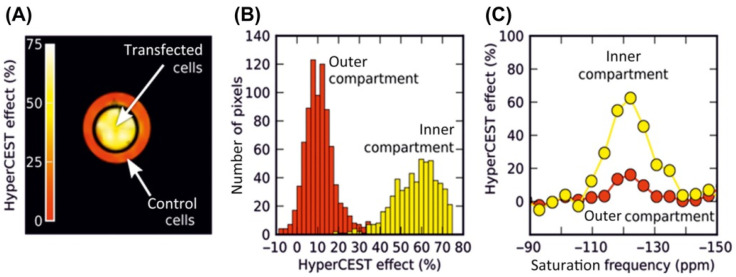
Claudin-4-targeted xenon HyperCEST imaging. (**A**) HyperCEST image (false colored) overlaid with a high-resolution ^1^H MRI scan for reference (black and white in the background to define the geometry outline of the compartments for orientation). (**B**) The histogram of the HyperCEST effect. (**C**) Localized HyperCEST spectrum for each of the compartments. Adapted from Piontek et al. (2017) [136] with permission provided by John Wiley and Sons and Copyright Clearance Center.

**Figure 6 ijms-25-01939-f006:**
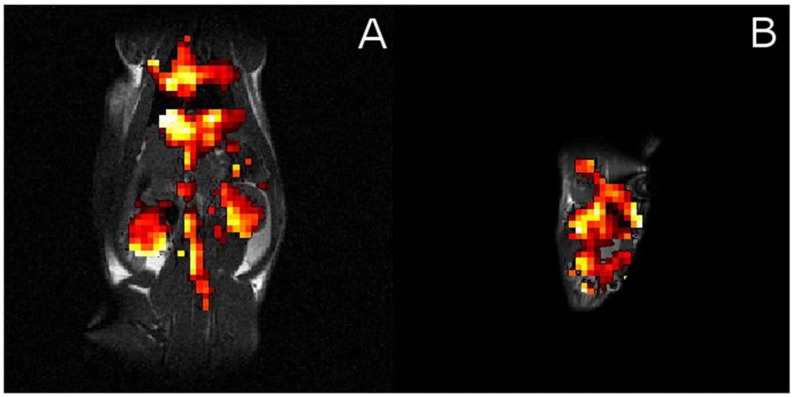
HyperCEST saturation maps of a (**A**) rat abdomen and (**B**) rat head, acquired after injection of 10 mM CB6 in PBS. Reproduced from Hane et al. (2017) [100] under a CC BY 4.0 DEED License.

**Figure 7 ijms-25-01939-f007:**
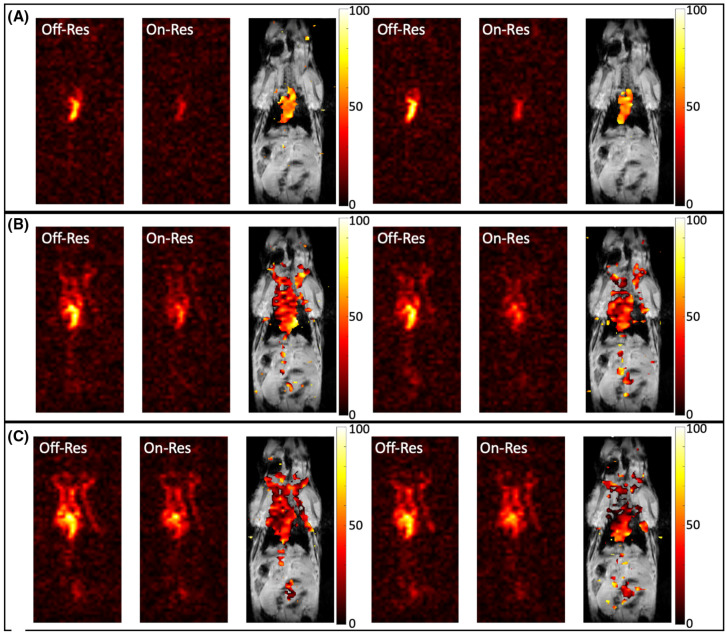
*In vivo* HyperCEST maps (two sets) obtained after the injection of 0.1 mL 20 mM CB6 in PBS solution into mice following (**A**) one breath, (**B**) three breaths, and (**C**) four breaths of HP ^129^Xe. Reproduced from McHugh et al. (2022) [102] with permission from John Wiley and Sons and Copyright Clearance Center.

**Figure 8 ijms-25-01939-f008:**
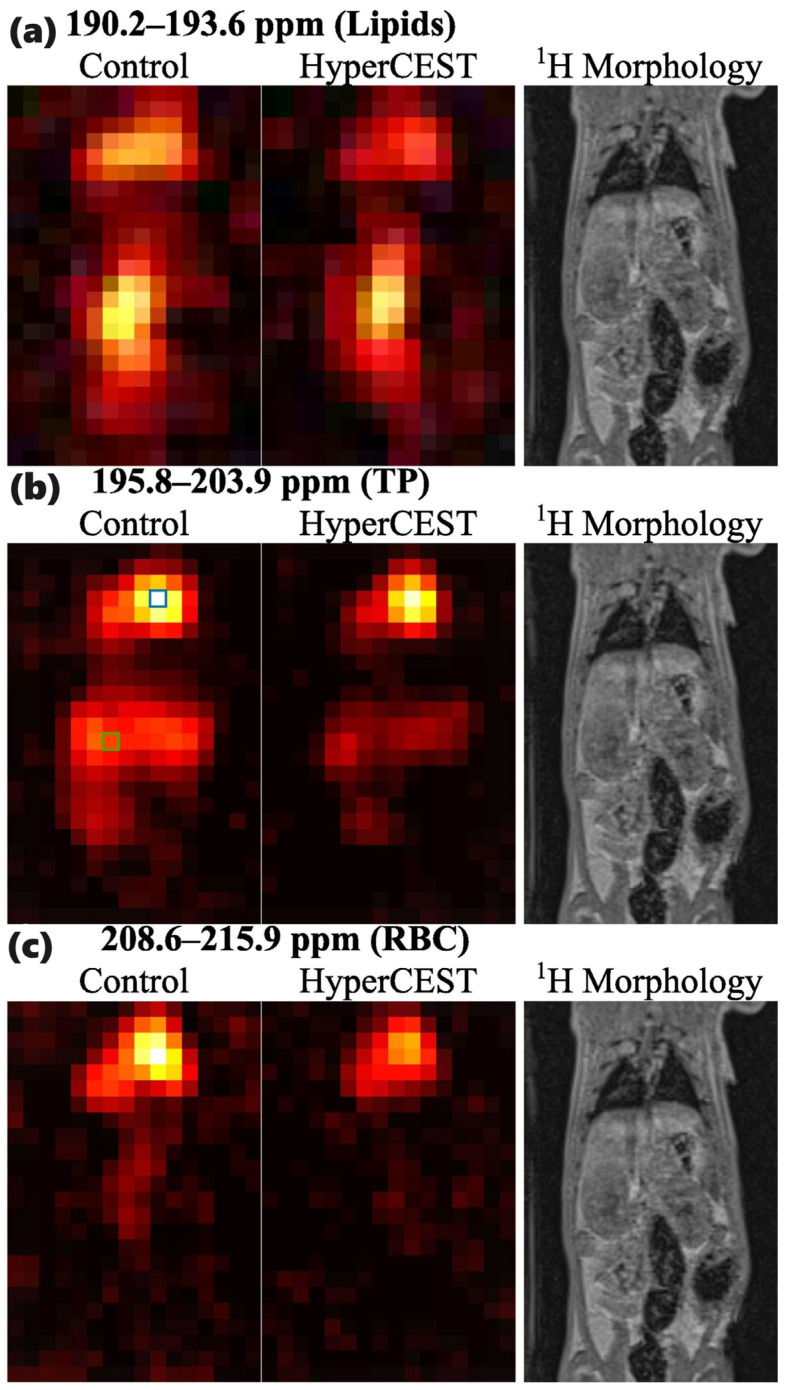
Maximum intensity projection of magnitude CSI data from HP ^129^Xe in (**a**) lipids, (**b**) aqueous tissues, and (**c**) red blood cells. The HyperCEST data were acquired after the injection of a 5 mM CB6 solution in saline. Reproduced from Kern et al. (2023) [104] under a CC BY 4.0 DEED License.

**Figure 9 ijms-25-01939-f009:**
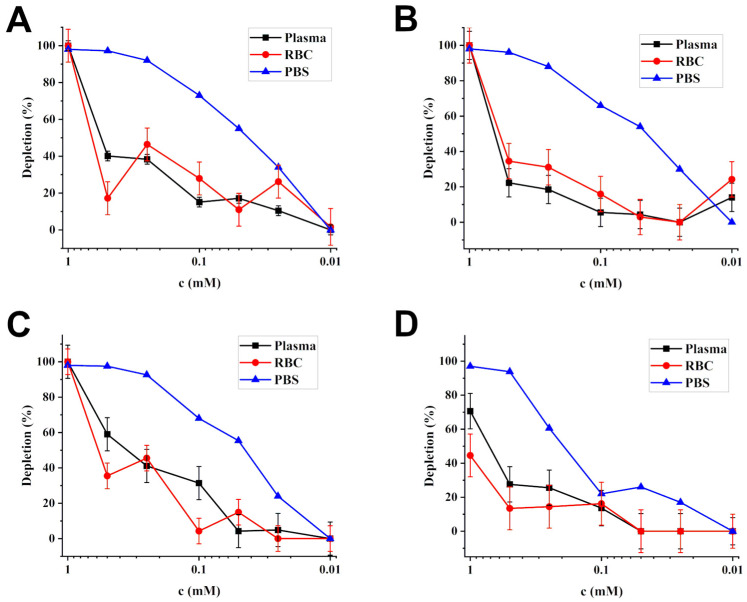
HyperCEST performance of (**A**) sinusoidal, (**B**) block, (**C**) 3-lobe sinc, (**D**) hyperbolic secant depolarization pulses over different concentrations of CB6 in bovine blood and PBS. Reproduced from Grynko et al. (2023) [103] under a CC BY 4.0 DEED License.

**Figure 10 ijms-25-01939-f010:**
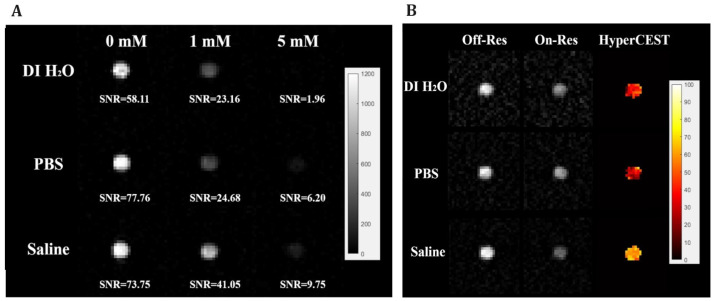
(**A**) HP ^129^Xe in control, 1 mM, and 5 mM of R3-Noria-MeSO_3_H and (**B**) HyperCEST images in 1 mM of R3-Noria-MeSO_3_H for DI H_2_O, PBS, and saline solutions. Reprinted with permission from Shepelytskyi et al. (2023) [204]. Copyright 2023 American Chemical Society.

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
