# Peer review of "Hyperpolarized Xenon-129 Chemical Exchange Saturation Transfer (HyperCEST) Molecular Imaging: Achievements and Future Challenges"

_ijms, 2024, doi:10.3390/ijms25031939_

Round 1
Reviewer 1 Report
Comments and Suggestions for Authors
I really enjoyed reading this review by Batarchuk et al. The manuscript contains a lot of interesting and well-structured information on the important topic of molecular imaging using Hyper-CEST. The quality of the presentation and the literature scope are good. I recommend publishing this review in its current form.
Author Response
Thank you very much for taking the time to review this manuscript. We highly appreciate your feedback.
Reviewer 2 Report
Comments and Suggestions for Authors
The manuscript “Hyperpolarized Xenon-129 Chemical Exchange Saturation Transfer (HyperCEST) Molecular Imaging: Achievements and Future Challenges” [ijms-2829186] written by Viktoriia Batarchuk, Yurii Shepelytskyi, Vira Grynko, Antal H. Kovacs, Aaron Hodgson, Karla Rodriguez, Ruba Aldossary, Tanu Talwar, Carson Hasselbrink, Iulian C. Ruset, Brenton DeBoef, Mitchell S. Albertis a review article about the application of hyperpolarized 129Xe biosensors in molecular magnetic resonance imaging.
The reviewer has expertise in the field of molecular structure determination organic chemistry, mainly by NMR and hence refers to the manuscript from this point of view.
The authors examine this topic comprehensively, but focus on the current developments of the last few years, in which this field has developed significantly. The different aspects from theoretical basics via implementation of equipment to applications and their current limitations are discussed. The main focus is on an understandable presentation of the physical and molecular principles, whereby hyperpolarization and the use of biosensors (complexing agents) are introduced. Furthermore, the implementation and representative results are impressively compiled. The presentation is precise, but still has a breadth and intensity that makes the entire topic understandable.
The entire review is clearly structured, comprehensibly written and covers the edited topic in full. All interesting and important aspects are thoroughly and critically addressed and discussed. The literature used is complete and no weight issues known to the reviewer have been ignored. From a content perspective, there are no complaints. The present work is hence of interest in the fields MRI, NMR, Spectroscopy, Biomedicine and Molecular Sciences. It is worth publishing in “International Journal of Molecular Sciences”. However, some details in the presentation of the manuscript might be improved. Hence there are some comments listed below, which should be taken into account by the authors prior to acceptance by “International Journal of Molecular Sciences”:
Comments:
1) Overall, the article is relatively long and has a clear structure. However, this does not go into very detailed subsections. Authors should therefore consider adding additional subheadings to make the structure of the article more clear.
2) In a few places the authors are not very precise, which may lead to misconceptions among readers. E.g. line 135: "polarization of the noble gas nuclei" refers to spin-polarization. The authors are therefore encouraged to examine any simplifying formulations they may have used and to clarify them if necessary.
3) Figure 1 impressively illustrates spin-exchange optical pumping. However, the legend of the figure seems a little short, especially for readers who are unfamiliar with the subject. The authors are therefore encouraged to expand on the legend and explain the character better to the reader. Also at the risk of creating redundancies with the overall text.
Minor Comments:
3) Lines 78 and 1348 “Schröder” instead of “Schroder”.
4) Relaxation time “T1”, “T2” with “T” in italics.
5) Lines 454-455: "It was shown[166] that the identical Xe@cryptophane-A 454 monoacid (CrA-ma)." The reviewer does not recognize any real message in this sentence.
6) Line 595 (and elsewhere?): The authors should consistently use the corresponding abbreviations of the SI units: here "h" instead of "hours".
7) Line 799 "p-benzenesulfomamide" should be "p-benzenesulfonamide" with "n" instead of "m" as well as "p" in italics.
8) E.g. "(ex vivo)" in line 1050. The authors should use "in vivo" and "ex vivo" in italics throughout.
Author Response
Thank you very much for taking the time to review this manuscript. Please see the attachment.
